# Myosin efflux promotes cell elongation to coordinate chromosome segregation with cell cleavage

Emilie Montembault[1], Marie-Charlotte Claverie[1], Lou Bouit[1], Cedric Landmann[1], James Jenkins[1], Anna Tsankova[2], Clemens Cabernard[2] & Anne Royou[1]

Chromatid segregation must be coordinated with cytokinesis to preserve genomic stability. Here we report that cells clear trailing chromatids from the cleavage site by undergoing two phases of cell elongation. The first phase relies on the assembly of a wide contractile ring. The second phase requires the activity of a pool of myosin that flows from the ring and enriches the nascent daughter cell cortices. This myosin efflux is a novel feature of cytokinesis and its duration is coupled to nuclear envelope reassembly and the nuclear sequestration of the Rho-GEF Pebble. Trailing chromatids induce a delay in nuclear envelope reassembly concomitant with prolonged cortical myosin activity, thus providing forces for the second elongation. We propose that the modulation of cortical myosin dynamics is part of the cellular response triggered by a "chromatid separation checkpoint" that delays nuclear envelope reassembly and, consequently, Pebble nuclear sequestration when trailing chromatids are present at the midzone.

[1] University of Bordeaux, CNRS, UMR5095, Institut Européen de Chimie et Biologie, 2 Rue Robert Escarpit, Pessac 33607, France. [2] Department of Biology, University of Washington, Seattle, WA 98195, USA. Correspondence and requests for materials should be addressed to E.M. (email: e.montembault@iecb.u-bordeaux.fr) or to A.R. (email: a.royou@iecb.u-bordeaux.fr)

Mitosis is the process by which the genome is transmitted from a mother cell into two daughter cells. Mitosis can be sub-defined into two phases: mitotic entry and mitotic exit. During mitotic entry in animal cells, microtubules rearrange into a bipolar spindle and chromatin condenses into distinct chromosomes concomitantly with the breakdown of the nuclear envelope. Mitotic entry culminates at metaphase when all the chromosomes are properly attached to the spindle. Subsequent mitotic exit ends when the two daughter cells have inherited a set of chromatids and the two cells physically separate. An elaborately ordered set of events define mitotic exit commencing with the separation of sister chromatids and their segregation toward each pole at anaphase. When the chromatids have reached the poles, chromatin decondensation ensues

**Figure 1** The presence of trailing chromatids at the midzone triggers the assembly of a wide contractile ring. **a** Myosin dynamics in cells carrying normal-length chromatid arms (NC) and cells with trailing chromatid arms (TC). Time-lapse images of live Drosophila third instar larvae neuroblasts expressing a chromatin marker, H2Az::mRFP (His, *red*) and a non-muscle myosin II marker, Sqh::GFP (Myo, *gray*). Neuroblasts divide asymmetrically to give rise to two daughter cells of different sizes, a large neuroblast (*Nb*) and a small ganglion mother cell (*GMC*) (*white dashed lines*). See Supplementary Movie 1. Time = min:s. *Scale bars* = 5 μm. **b** Images of two different cells with NC or TC expressing H2Az::mRFP (His, *red*) and Sqh::GFP (Myo, *gray*) at the onset of furrowing. The length of the contractile ring is indicated with *yellow brackets*. The *cyan arrows* point to TC. *Scale bars* = 5 μm. **c** Scheme of a cell at onset of furrowing to illustrate the measurements of ring length and total cell length. **d** Scatter dot plot showing the distribution of ring length at the onset of furrowing. **e** Graph showing the linear correlation ($R^2$ = 0.66) of the total cell length with the ring length at the onset of furrowing. **f, g** Scatter dot plots showing the integrated density (**f**) and mean intensity (**g**) of myosin signal at the contractile ring of cells with NC and cells with TC specifically exhibiting wide rings at the onset of furrowing. **h** Graph showing the diameter of the contractile ring over time for cells with NC or TC with wide rings from AO. Data points were fit to a sigmoid curve. The Hillslope (*h*) is used to compare the rate of furrow invagination. All cells are oriented so that the daughter Nb is apical and the daughter GMC is basal. Time 0:00 corresponds to anaphase onset (AO, initiation of sister chromatids separation). *n* = number of cells. The mean ± 95% CI is presented for all graphs and scatter dot plots. A Mann-Whitney test was used to calculate *P* values (**** corresponds to *P* < 0.0001)

concomitantly with nuclear envelope reassembly during telophase. Meanwhile, cytokinesis, the process of cell cleavage occurs. Signals from the central spindle, an anti-parallel bundle of microtubules that are organized between the two chromatin masses, define the cleavage site[1]. The centralspindlin complex composed of MgcRacGAP/RacGAP50c and MKLP1/Pavarotti drives the localization of the guanine exchange factor for RhoA (RhoGEF) (called Pebble in *Drosophila*) at the midzone of the central spindle, which, in turn, activates RhoA (Rho1 in *Drosophila*) (reviewed in ref. [2]). Activation of Rho1 mediates the assembly of the contractile ring, composed of actin and non-muscle myosin II (myosin) filaments that are anchored to the plasma membrane by anillin[3–5]. The constriction of the ring triggers the invagination of the plasma membrane that forms the cleavage furrow. Abscission terminates cytokinesis by pinching off the plasma membrane thereby physically separating the daughter cells.

Coordination of these mitotic exit events is essential to ensure successful mitosis. In particular, cells must avoid the entrapment of trailing chromatid (TC) arms by the cytokinetic ring. In most animal cells, chromosomes are metacentric, where their centromeres are located on one small region of the sister chromatids. As a result, when the kinetochore pulls the sister chromatids poleward, the centromere reaches the pole first, while the

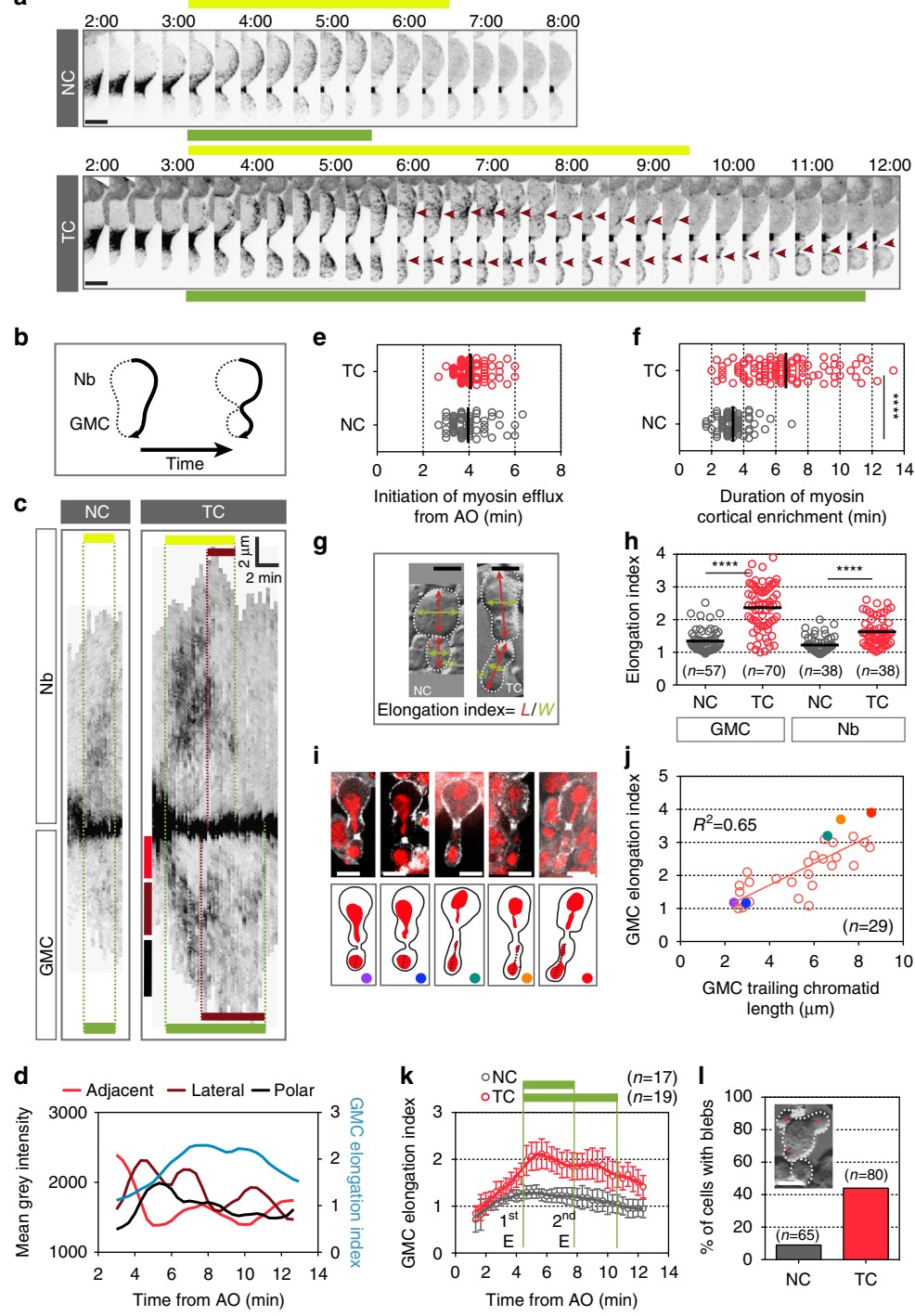

chromatid arms trail. This scenario necessitates mechanisms that coordinate the clearance of TCs from the cleavage site with the completion of ring closure. Recent studies have revealed such mechanisms. In budding yeast, chromatids that remain at the midzone are thought to trigger an aurora B-mediated "no cut" pathway that delays abscission[6, 7]. A similar abscission checkpoint is found in human cells in the presence of chromatin bridges[8–11]. Other studies have monitored the kinetics of cell division in cells with long chromatid arms. While plant cells have difficulties coordinating cell division with the segregation of large chromosomes[12, 13], fungi and insect cells have evolved different mechanisms to ensure successful cytokinesis in the presence of TCs. In budding yeast, long chromatid arms undergo prolonged condensation to clear the midzone before abscission[14]. Drosophila neuronal stem cells transiently elongate to adapt to the length of the TC arms during their segregation. This cell elongation involves a change in myosin dynamics during cytokinesis and requires the activity of the RhoGEF Pebble (Pbl)[15]. However, the mechanism underlying myosin-mediated cell elongation has not been fully elucidated. In this study, we investigate myosin dynamics during cytokinesis and its role during cell elongation in the presence of TCs. We report that during contractile ring ingression, a pool of myosin undergoes efflux from the ring toward the poles. This transient cortical enrichment of myosin is a novel and conserved feature of cytokinesis that is mediated by the Rho-GEF Pebble. The presence of TCs prolongs myosin cortical enrichment concomitant with a delay in nuclear envelope reassembly and nuclear sequestration of Pebble. During this extended cortical activation, myosin reorganizes into broad lateral rings, which contraction promotes cell elongation, thus facilitating the clearance of TC arms from the cleavage plane. Finally, we show that the cytoplasmic retention of Pebble is sufficient to maintain myosin cortical activity and promote cell elongation in the absence of TCs.

## Results

**Effect of TCs on myosin ring assembly**. To decipher the mechanisms underlying cell elongation in the presence of TCs, we monitored the dynamics of myosin using its regulatory light chain (encoded by *spaghetti squash* gene, sqh) fused to GFP or RFP during cytokinesis in Drosophila larval neuroblasts. The neuroblast divides asymmetrically to give rise to a neuroblast (Nb) and a ganglion mother cell (GMC). We compared cells with TC arms to cells with normal chromosomes (NC) (see "Methods" section). One to two minutes after the initiation of sister chromatid separation, which defines anaphase onset, myosin depleted the poles and accumulated at the presumptive cleavage site to form the contractile ring in both cell types (Fig. 1a, Supplementary Figs. 1a and 2a–b, and Supplementary Movie 1). At the onset of furrowing, most cells with TC exhibited a wider myosin ring, correlated with a mild increase in total cell length (Fig. 1b–e and Supplementary Fig. 2b). In addition, the rate at which the central band of myosin collapses to a ring was delayed in cells with TC (Supplementary Fig. 1b). Quantitative analysis of myosin signal at the ring at furrowing onset revealed an overall increase in the amount of myosin during the assembly of wide rings in cells with TC (Fig. 1f), while the average myosin signal at the ring was not affected (Fig. 1g). This suggests an active enrichment of myosin during ring assembly when chromatids remain at the midzone. The assembly of a wide ring subsequently mildly affected the rate of furrow invagination (Fig. 1h).

**Prolonged cortical myosin enrichment in the presence of TC**. Strikingly, at mid-cytokinesis, a pool of myosin initiated outward flow from the contractile ring and invaded the cortex of both nascent cells (Figs. 1a and 2a, c, Supplementary Fig. 2a, and Supplementary Movie 1). We confirmed that these cortical myosin dynamics were not due to overexpression of Sqh by monitoring one copy of Sqh::GFP in a $sqh^{AX3}$ null-mutant cells with NC, which exhibited similar patterns (Supplementary Fig. 3a). Importantly, transient myosin cortical enrichment was observed in wild-type embryonic and pupal epithelial dividing cells, indicating that myosin efflux is a common feature of cytokinesis (Supplementary Fig. 3b). In cells with NC, this cortical myosin enrichment persisted for 3 min, on average, after efflux initiation and correlated with a slight elongation of both daughter cells (Fig. 2f–h, k, NC elongation index >1). Then, myosin rapidly

**Figure 2** Myosin transiently decorates the cortex of the nascent daughter cells and reorganizes into lateral rings in the presence of trailing chromatids. **a** Time-lapse images of cells with NC or TC expressing Sqh::GFP during cytokinesis. Half of the cell cortex is shown starting 2 min after anaphase onset. At mid cytokinesis, myosin undergoes outward flow from the contractile ring toward the polar cortex (called efflux) in NC and TC cells. The duration of myosin enrichment at Nb and GMC cortices are illustrated above (*yellow line*) and below (*green line*) the images, respectively. The *brown arrowheads* point to the reorganization of myosin into broad lateral rings, whose constriction forms pseudo cleavage furrows. Time = min:s. **b** Scheme showing the method for generating kymographs in Fig. 2c. **c** Kymographs of cortical myosin for cells with NC or TC presented in Fig. 2a. The *yellow* and *green horizontal bars* represent the duration of myosin efflux from initiation to disappearance in Nb and GMC, respectively. The *brown horizontal bars* correspond to the time of contraction of the lateral ectopic myosin rings. The *red*, *brown*, and *black vertical bars* designate the regions where the mean gray intensity of the myosin signal is measured and plotted in Fig. 2d. The *x* and *y* axes *scale bars* correspond to 2 min and 2 μm, respectively. **d** Graph showing the average myosin level at three distinct cortical regions (adjacent to the contractile ring (*red curve*), lateral (*brown curve*), and polar (*black curve*)) of the GMC over time (left *y* axis). The GMC elongation index (*cyan curve*) is indicated in the same graph (right *y* axis). **e** Scatter dot plot showing the time of myosin efflux initiation from anaphase onset. (NC, n = 57; TC, n = 84). **f** Scatter dot plot showing the duration of myosin cortical enrichment from initiation of efflux to disappearance. (NC, n = 69; TC, n = 100). **g** DIC images of cells with NC and TC to illustrate the calculation of the elongation index plotted in Fig. 2h, j, k, which correspond to the ratio: length (*L*) (*red double arrow*) over width (*W*) (*green double arrow*) of each daughter cell. **h** Scatter dot plot showing the distribution of the elongation index of the GMC and Nb daughter cells with NC or TC. **i** Still images of live cells expressing H2Az::mRFP (*red*) and Sqh::GFP (*gray*) with TC and the corresponding scheme below. These cells are examples of cells where the length of the trailing chromatid was measured (*black dashed line*) at the time the elongation index was calculated and plotted in Fig. 2j. The *colored dots* correspond to the *colored dots* plotted in Fig. 2j. **j** Graph showing the correlation between the length of the TC and the elongation index for the GMC. The *colored dots* correspond to the cells shown in Fig. 2i. **k** Graph showing the GMC elongation index plotted over time for NC and TC that illustrates the first (1st E) and second elongation (2nd E). The *green horizontal bars* above the graph correspond to the average duration of myosin cortical enrichment for NC and TC, respectively. The *vertical green dotted lines* correspond to the average initiation of efflux and disassembly of myosin from the cortex. **l** Frequency of cells with NC and TC that bleb. The DIC image illustrates a cell with blebs (*red stars*). All cells are oriented so that the daughter Nb is apical and the daughter GMC is basal. *Scale bars* = 5 μm. n = number of cells. The mean ± 95% CI is presented for all graphs and scatter dot plots. A Mann-Whitney test was used to calculate *P* values (**** corresponds to *P* < 0.0001)

disassembled from the cortex, with the exception of the midbody (Figs. 1a and 2a, c). In cells with TC, myosin initiated efflux at a similar time after anaphase onset as in control cells (Fig. 2e). However, the period of myosin cortical enrichment was greatly prolonged (Fig. 2f and Supplementary Fig. 2c). After propagating toward the polar cortex, myosin depleted the area adjacent to the contractile ring, and to some extent the poles, and accumulated on the lateral cortex to form broad rings (Fig. 2a, c, d). In some instances, the contraction of these broad lateral rings induced the invagination of pseudo-cleavage furrows (Fig. 2a, c, *brown arrowheads* and *bars*, respectively, and Supplementary Movie 1), resulting in the dramatic elongation of both nascent daughter cells, which was evident by measuring the length-to-width ratio (Fig. 2d, *cyan curve*, Fig. 2g, h, k, and Supplementary Fig. 2d). As observed previously, the extent of cell elongation

was correlated with the length of the TC and was accompanied by a mild elongation of the spindle without any changes in the rate of chromosomes segregation (Fig. 2i, j and Supplementary Fig. 4a–f)[15]. Finally, cells with TC frequently exhibited blebs at the poles or near the contractile ring during cell elongation (Fig. 2l).

**Myosin undergoes efflux from the contractile ring.** To determine the origin of the myosin pool that transiently invades the cortex during ring constriction, we carefully monitored myosin dynamics every second at the time myosin initiated efflux in cells with normal chromatids. We observed that a pool of myosin invaded the area adjacent to the ring in both nascent cells. Myosin signal progressed to the lateral region of the cortex and finally

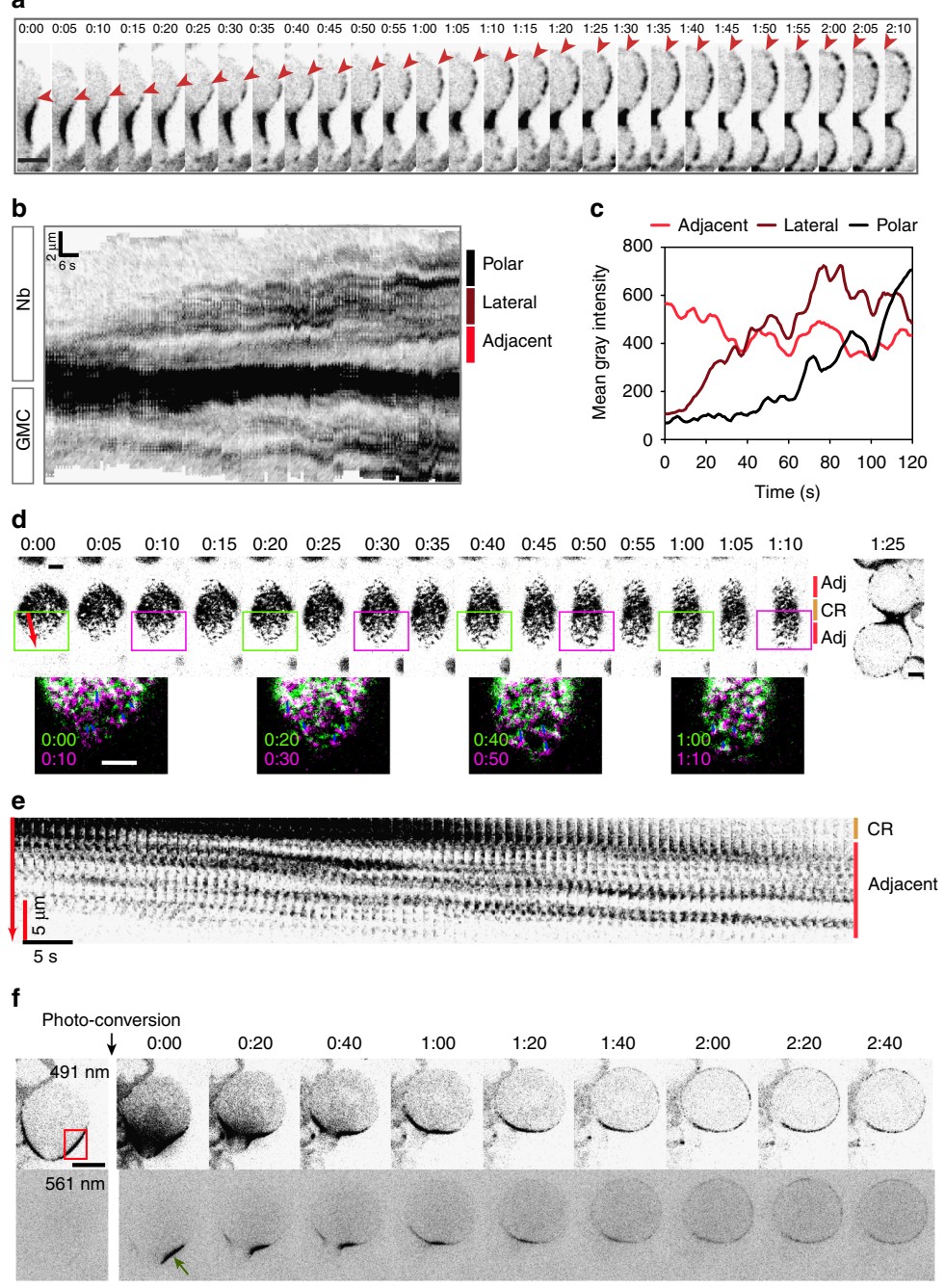

propagated to the poles within 2 min after efflux initiation (Fig. 3a, *red arrowheads*, Fig. 3b, c, and Supplementary Movie 2). Examination of the surface of the cortex during efflux revealed discrete patches of myosin being extruded from the ring and moving diagonally toward the pole (Fig. 3d, e). To confirm that the pool of myosin that invades the cortex originates from the contractile ring, we monitored dividing cells expressing Sqh fused to Dendra (Sqh::Dendra), enabling the photoconversion of a defined pool of myosin at a specific time during cytokinesis. When a cytoplasmic pool of Sqh::Dendra was photo-converted 2 min after anaphase onset, no visible Sqh::Dendra signal was detected at the cortex at the time of myosin efflux (Supplementary Fig. 5a). In contrast, when Sqh::Dendra was photo-converted at the site of the contractile ring, the photo-converted Sqh::Dendra signal was visible at the cortex a few minute later (Fig. 3f and Supplementary Fig. 5b). Similar results were observed in cells with TCs (Supplementary Fig. 5c). These data indicate that most of the pool of myosin that invades the whole cortex during cytokinesis originates from the contractile ring. In addition, our observations reveal that some myosin filaments undergo cortical flow from the ring toward the cortex. We can not exclude that a pool of cytoplasmic myosin contributes to the transient enrichment of myosin at the cortex.

**The RhoGEF Pebble mediates myosin efflux**. The assembly and constriction of the contractile ring requires Rho1 activity, which promotes the activation of Rho kinase (Rok). Rok, in turn, activates myosin by phosphorylating its regulatory light chain[16]. Rho1 activity is controlled by its guanine exchange factor Ect2 (Pebble (Pbl) in Drosophila)[17–21]. We have previously shown that cells with reduced Pbl activity (using $pbl^5$ and $pbl^{MS}$ hypomorphic alleles) fail to elongate to adapt to the presence of TCs at the midzone. As a result, *pbl* mutant flies exhibited severe cell loss during development[15]. Since myosin cortical enrichment is a prerequisite for cell elongation and since *pbl* mutant cells fail to elongate during the segregation of TCs, we investigated whether Pbl contributes to myosin cortical enrichment. To do so, we monitored myosin dynamics in the $pbl^3$ (null allele) or $pbl^5$ over $pbl^{MS}$ mutants (called *pbl* mutant) that reduce Pbl activity without compromising cytokinesis in 90% of cells[15]. Cells with reduced Pbl activity assembled a contractile ring that was narrower than those observed in wild-type cells (Fig. 4a–c). In addition, no myosin efflux or cortical enrichment was detected in two-thirds of *pbl* cells regardless of the presence of TCs (Fig. 4d, e and Supplementary Movie 3). In *pbl* mutant cells with TC where myosin efflux was observed, the time of myosin cortical

enrichment was brief (Fig. 4f). Consequently, *pbl* mutant cells failed to elongate in the presence of TCs (Fig. 4a, g-i and Supplementary Movie 3). Our data indicate, first, that Pbl and hence Rho1 activity is required for myosin efflux and, second, that Pbl-dependent myosin contractile forces drives cell elongation.

**Delay in nuclear envelope reassembly in the presence of TC**. We uncovered a specific myosin behavior, myosin efflux, that occurs during anaphase and seems inherent to the constriction of the contractile ring. The timing of myosin cortical enrichment (from efflux initiation to complete disassembly of myosin from the cortex) is greatly prolonged in the presence of TCs (Fig. 2f). This suggests a surveillance mechanism that controls the dynamics of cortical components when chromatids are present at the cleavage site. The notion of a "chromosome separation checkpoint" has been proposed previously for the coordination of chromosome separation with nuclear envelope reassembly (NER)[22, 23]. The authors uncovered an Aurora B-dependent delay in NER when broken chromosomes are present at the midzone during anaphase.

Thus, we investigated whether NER was delayed during the segregation of TCs. We monitored the dynamics of the nuclear pore component Nup107 fused to GFP in cells with NC and TC. In control cells, GFP::Nup107 accumulated at the apex of the segregated chromosome mass on average 3 min after anaphase onset in both nascent cells (Fig. 5a, b, initiation of NER and Supplementary Movie 4). GFP::Nup107 surrounded the entire chromosome mass within 3 min following its first appearance, thus defining the completion of NER (Fig. 5a, c). In the presence of TC, the time of appearance of GFP::Nup107 on the chromosome mass was slightly delayed (Fig. 5a, b and Supplementary Movie 4). In addition, we observed that the completion of NER was severely delayed in the majority of cells with TC. Consistent with previous findings, the TC arm was specifically devoid of GFP::Nup107 signal until it had reached the chromosome mass at the pole (Fig. 5a, *white arrowheads* and Supplementary Movie 4)[22, 24]. This delay in NER was borne out by the observation that a GFP::NLS (Nuclear Localization Signal) probe accumulated in the nucleus at a slower rate in cells with TC compared to control cells (Fig. 5d–f).

Thus, the presence of TCs at the midzone is associated with a delay in both the completion of NER and the timing of myosin dissociation from the whole cortex. Next, we determined whether the duration of myosin efflux was correlated with the timing of NER. To do so, we examined the dynamics of both myosin and

**Figure 3** A pool of myosin undergoes efflux from the contractile ring and enriches the daughter cell cortices during cytokinesis. **a** Sagittal view of a cell with NC expressing Sqh::GFP. Half of the cell cortex is presented at the time of myosin efflux initiation (Time 0:00). The *red arrowheads* indicate the edge of the myosin signal propagating in the nascent Nb cortex. Myosin reaches the Nb pole within 2 min. See Supplementary Movie 2. **b** Kymograph of cortical myosin of the cell presented in **a** showing Sqh::GFP dynamics from one pole to the other as described in Fig. 2b and method section. The *x* and *y* axes *scale bars* indicate 6 s and 2 μm, respectively. The *red*, *brown*, and *black vertical bars* designate the regions where the mean gray intensity of the myosin signal is measured and plotted in **c**. **c** Graph showing the mean gray intensity of myosin in three regions (adjacent to the contractile ring (*red*), lateral (*brown*), and polar (*black*)) of the Nb cortex over time. Time = 0 corresponds to the time of myosin efflux initiation. **d** Surface view of a sqh::GFP labeled cell with NC showing the contractile ring (CR, *orange vertical bar*) and the regions adjacent to the ring (adj, *red vertical bars*) at the time of myosin efflux initiation. The last image is a sagittal view of the same cell at 1 min25s. The *red arrow* represents the region used for the kymograph in **e**. The *green* and *magenta rectangles* correspond to the *insets* below. The *insets* show the overlay of two time points of 10 s interval (early time point in *green*, late time point in *magenta*). *Blue arrows* represent the displacement of some myosin patches within 10 s. **e** Kymograph of the cell shown in **d**. Patches of myosin move out of the contractile ring (CR, *orange vertical bar*) and diffuse along the region adjacent to the ring (*red vertical bar*) toward the pole. The *x* and *y* axes *scale bars* correspond to 5 s and 5 μm, respectively. **f** Time lapse of a cell with NC expressing Sqh::Dendra2. Dendra2 undergoes *green* to *red* photoconversion upon absorption of 405 nm light. No signal is detected with a 561 nm excitation wavelength before photoconversion. The *red rectangle* corresponds to the area photoconverted with the 405 nm laser. Upon irradiation at the site of the contractile ring (time 0:00), the pool of photoconverted Sqh::Dendra2 is rapidly visible with the 561 nm laser (*green arrow*). Two minutes after irradiation, the pool of photoconverted Sqh::Dendra2 is detected around the cortex similar to the Sqh::Dendra2 detected with the 491 nm laser. Time = min:s. *Scale bars* = 5 μm

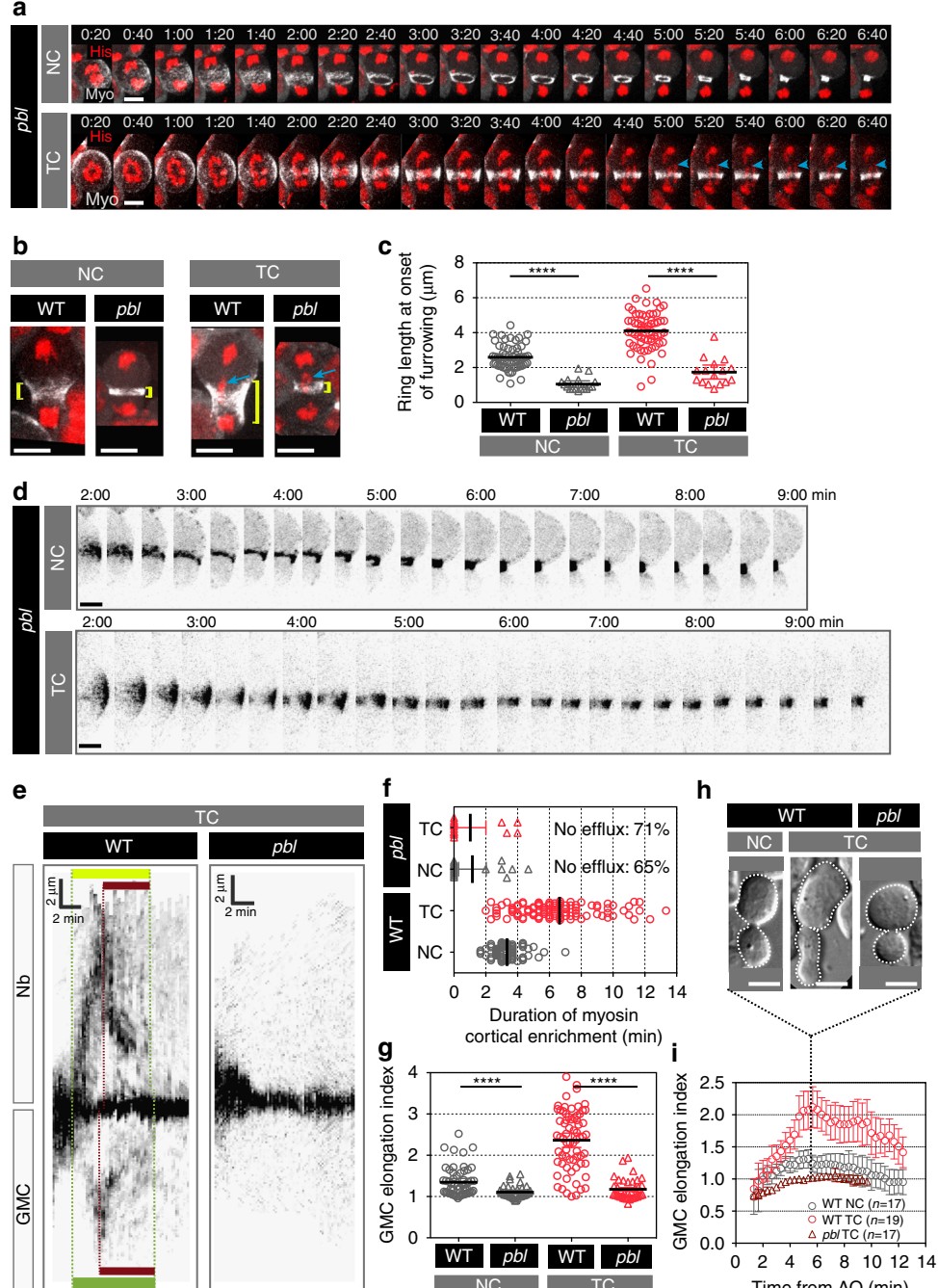

**Figure 4** The RhoGEF Pebble mediates myosin efflux. **a** Time-lapse of pebble (*pbl*) mutant cells expressing H2Az::mRFP (His, *red*) and Sqh::GFP (Myo, *gray*) with NC or TC. The *blue arrowheads* indicate trailing chromatid arms. See Supplementary Movie 3. **b** Images of wild type (*WT*) and *pbl* mutant cells with NC or TC expressing H2Az::mRFP (*red*) and Sqh::GFP (Myo, *gray*) from time-lapse movies at the onset of furrowing. The length of the contractile ring is indicated with *yellow brackets*. The *blue arrows* point to TC arms. *Scale bars* = 5 μm. **c** Scatter dot plot showing the distribution of the ring length at the onset of furrowing in WT (as shown in Fig. 1d) and *pbl* mutant cells with NC or TC. (*pbl* NC, *n* = 17; *pbl* TC, *n* = 16). **d** Time-lapse images of *pbl* mutant cells with NC or TC expressing Sqh::GFP during cytokinesis. Half of the cell cortex is shown starting 2 min after anaphase onset. **e** Kymographs of WT or *pbl* mutant cells with TC showing Sqh::GFP dynamics from one pole to the other. The *yellow* and *green horizontal bars* show the duration of myosin cortical efflux from initiation to disappearance in Nb and GMC, respectively. The *brown bars* correspond to the time of lateral myosin ring contraction. The *scale bars* on the *x* and *y* axes indicate 2 min and 2 μm, respectively. **f** Scatter dot plot showing the duration of myosin efflux in WT cells with NC or TC (as shown in Fig. 2f) and *pbl* mutant cells with NC or TC. No myosin efflux is visible in more than 60% of *pbl* mutant cells regardless of the presence of trailing chromatid arms. (*pbl* NC, *n* = 17; *pbl* TC, *n* = 14). **g** Scatter dot plot showing the GMC elongation index of WT with NC or TC (as shown in Fig. 2h) and *pbl* mutant cells with NC or TC (*pbl* NC, *n* = 37; *pbl* TC, *n* = 38). **h** DIC images of WT and *pbl* mutant cells at 6 min after anaphase onset to illustrate the elongation of the nascent daughter cells associated with myosin efflux. The *white dashed lines* outline the cells. **i** Graph showing the GMC elongation index over time in WT (as shown in Fig. 2k) and *pbl* mutant cells. Time 0:00 corresponds to anaphase onset (AO). Time = min:s. *n* = number of cells. *Scale bars* = 5 μm. The mean ± 95% CI is presented for all graphs and scatter dot plots. A Mann-Whitney test was used to calculate *P* values (**** corresponds to *P* < 0.0001)

GFP::NLS in cells with NC or TC. Interestingly, in both cell types, myosin disappeared from the cortex at the time GFP::NLS started being enriched in the nucleus, which was severely delayed in cells with TC (Supplementary Fig. 6a–c). Consistently, the duration of myosin efflux correlated with the duration of NER in both cell types (Supplementary Fig. 6d). Collectively, these results reveal an

adaptive response to the presence of chromatids at the cleavage site that delays NER and prolongs cortical myosin activity.

**Cytoplasmic retention of Pbl prolongs cortical myosin activity.** Our results indicate that the duration of myosin efflux is coupled to the timing of NER. It may be that the reassembly of a

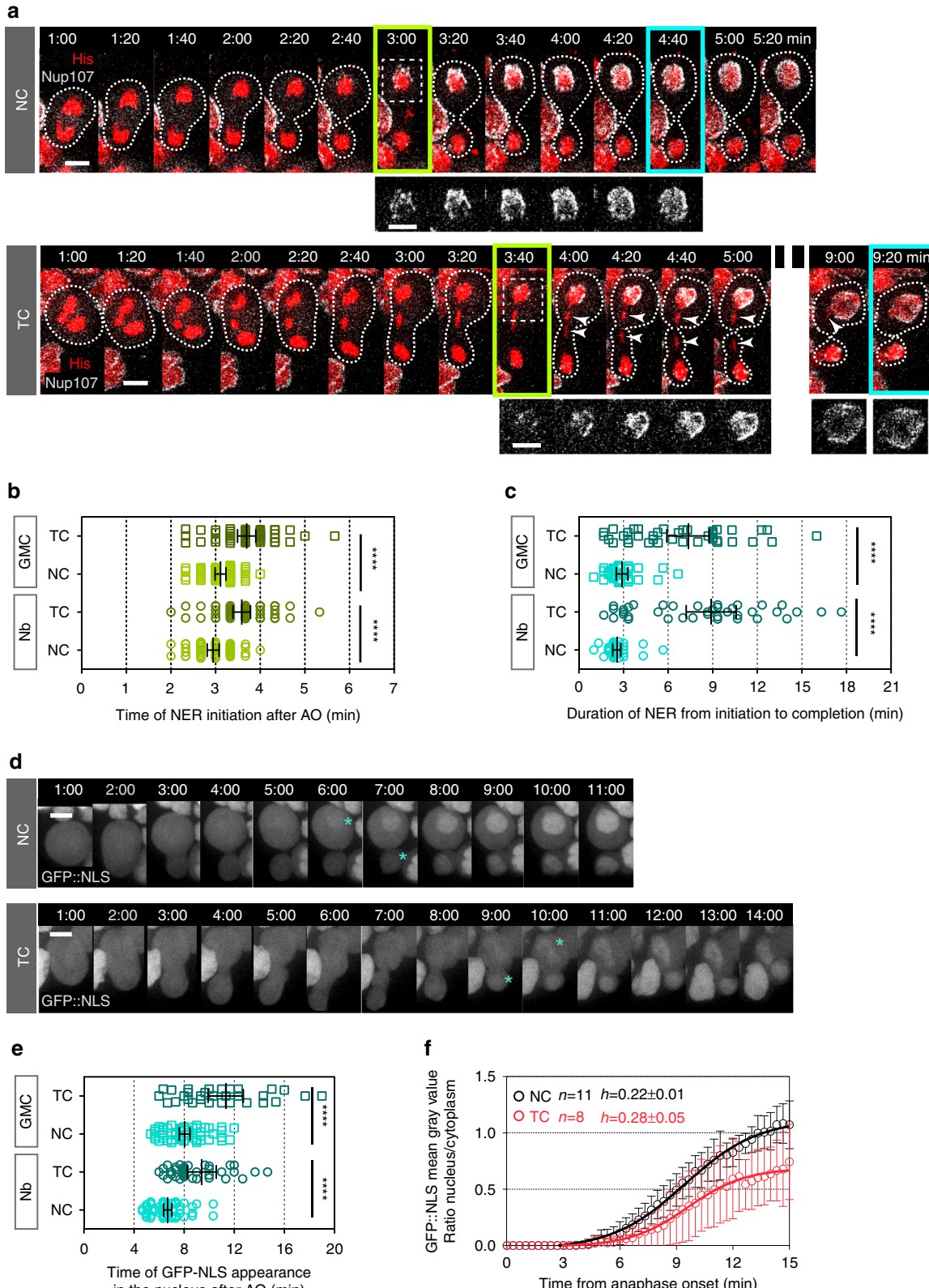

functional nuclear envelope promotes the nuclear sequestration of components controlling myosin cortical localization. Their subsequent depletion from the cytoplasm induces the dissociation of myosin from the cortex except for the midbody. Interestingly, Ect2/Pbl accumulates progressively in the nucleus at telophase and remains nuclear throughout interphase[25, 26]. Moreover, its release from the nucleus is required to trigger Rho1-dependent myosin cortical activity upon entry into mitosis[27, 28]. These observations associated with the fact that depletion of Pbl activity prevented myosin efflux and adaptive cell elongation led us to posit that the threshold of Pbl required to maintain myosin activity at the non-equatorial cortex may be lost upon reassembly of a functional nuclear envelope and subsequent sequestration of Pbl in the nucleus. To test this hypothesis, we created transgenic *pbl* mutant flies expressing a version of Pbl in which its NLS was mutated (Pbl-NLSmut) (Supplementary Fig. 7a)[25, 26]. As a control, we used transgenic *pbl* mutant flies carrying a transgene containing the sequence of wild-type Pbl (Pbl full length, Pbl-FL) inserted at the same locus as the Pbl-NLSmut transgene. Immunostaining of larval tissues showed that Pbl-FL was nuclear in interphase salivary gland cells and Nbs. The Pbl-NLSmut construct was cytoplasmic in salivary gland cells and Nbs. However, a faint signal was detected in the Nb nucleus, suggesting that nuclear import of Pbl-NLSmut is impaired but not prevented in this cell type (Supplementary Fig. 7b).

We next monitored the dynamics of myosin in *pbl* mutant Nbs expressing Pbl-FL or Pbl-NLSmut. In cells expressing Pbl-FL, myosin efflux, albeit weak, was visible in both nascent cells (Fig. 6a, b, *yellow* and *green bars* and Fig. 6d). The time of myosin efflux was extended during the segregation of TCs (Fig. 6d), and was associated with blebbing, as observed in wild-type cells (Fig. 6f, see Fig. 2i for wild-type cells). Interestingly, the expression of Pbl-NLSmut in cells with NC was associated with a prolonged period of cortical myosin activity (Fig. 6a–d and Supplementary Movie 5). In some cells, myosin reorganized into broad ectopic rings during prolonged efflux (Fig. 6a, b, *white arrowheads and brown bars*, respectively, and Supplementary Movie 5). This was accompanied by an increase in daughter cell length even in the absence of TCs (Fig. 6e and Supplementary Movie 5). A significant proportion of Pbl-NLSmut cells with NC exhibited extensive blebbing similar to wild type and Pbl-FL cells with TC (Fig. 6c, f). In addition, myosin displayed a peculiar behavior during prolonged cortical activity in Pbl-NLSmut cells, in which it underwent pulses of assembly and disassembly at different positions on the cortex. No blebs preceded these myosin flashes (Fig. 6b, c). Collectively, these results indicate that the retention of Pbl in the cytoplasm during mitotic exit suffices to mediate prolonged cortical myosin activity in the

absence of TCs. This is accompanied by an increase in the frequency of cells elongating and blebbing.

## Discussion

We have previously shown that cells adapt to TCs by undergoing myosin-mediated cell elongation during cell division[15]. In this study, we find that cells undergo two phases of elongation. The first phase depends on the assembly and constriction of a wide contractile ring. The second phase depends on the duration of myosin cortical enrichment, which relies on a novel myosin behavior that we refer to as myosin efflux. At mid-ring closure, a pool of myosin propagates from the contractile ring to the poles. Myosin efflux occurs systematically during cytokinesis regardless of the presence of TCs and is mediated by Rho1-GEF signaling. In cells with normal chromatids, myosin decorates the cortex briefly before dissociating in concert with NER and ring closure. In the presence of TCs at the midzone, global NER is delayed. Consequently, the retention of Pbl in the cytoplasm prolongs myosin activity at the nascent daughter cell cortex. Cortical myosin reorganizes into broad lateral ectopic rings, which transient constriction promotes the second adaptive elongation, facilitating the clearance of TCs from the midzone. Finally, upon completion of NER, myosin dissociates from the cortex, allowing rounding of the daughter cells while preserving their size asymmetry (Fig. 7).

Our examination of myosin dynamics during adaptive cell elongation enabled us to uncover a novel feature of cytokinesis, myosin efflux, shared by at least three Drosophila cell types, Nbs, embryonic and pupal epithelial cells. Our data indicate that the pool of myosin that decorates the cortex originates largely from the contractile ring. The pool of photo-converted Sqh::Dendra at the contractile ring labeled the cortex for a few minutes post conversion. Moreover, our dissection of myosin dynamics at high temporal resolution (one second intervals) indicates that myosin patches slide along the cortex from the ring toward the pole. Numerous studies have reported cortical flows and waves of actin and myosin from the poles toward the equator during ring assembly[29–33]. However, to our knowledge, this is the first description of dynamic myosin flux from the ring to the poles during ring constriction. Given its brevity, myosin efflux might have been overlooked in other studies on myosin dynamics during cytokinesis. Moreover, some cells maintain a significant quantity of actin and myosin at the polar cortex, which may obscure the detection of subtle fluctuations of polar myosin during cytokinesis[34–36].

It is well established that Pbl-mediated Rho1 signaling promotes contractile ring assembly[17–21, 37]. Here we provide evidence that myosin efflux requires Pbl activity. First, no

**Figure 5** Initiation and completion of nuclear envelope reassembly are delayed in cells with trailing chromatids. **a** Time-lapse images of cells with NC or TC expressing H2Az::mRFP (His, *red*) and GFP::Nup107, a marker for nuclear pores (Nup107, *gray*). Images highlighted by a *green rectangle* correspond to the time of appearance of GFP::Nup107 around the mass of chromosomes in the daughter Nb, which defines the time of initiation of nuclear envelope reassembly (*NER*). Images highlighted with a *cyan rectangle* correspond to the time of completion of NER in the daughter Nb. The *white arrowheads* point to the trailing chromatid arms. The *white dotted lines* outline the cells. The *white dashed square* indicates the zone selected for the insets. The *insets* correspond to the GFP::Nup107 signal. See Supplementary Movie 4. **b** Scatter dot plot showing the time of NER initiation as defined in a in GMC and Nb daughter cells with NC ($n = 48$) or TC ($n = 56$). **c** Duration of NER from initiation to completion as defined in a in GMC and Nb daughter cells with NC ($n = 38$) or TC ($n = 41$). **d** Time-lapse images of cells with NC or TC expressing GFP::NLS starting 1 min after anaphase onset. *Cyan asterisks* indicate the time of appearance of the GFP::NLS signal in the nucleus of the GMC and Nb. **e** Scatter dot plot showing the time of appearance of GFP::NLS in the nucleus in GMC and Nb daughter cells with NC ($n = 59$) or TC ($n = 37$ and 32, respectively) after anaphase onset. The time of GFP::NLS nuclear accumulation is severely delayed in cells with TC compared to cells with NC. **f** Graph showing the ratio of nuclear/cytoplasmic average intensity of GFP::NLS signal, over time from anaphase onset in cells with NC or TC. Data points were fit to a sigmoid curve. The Hillslope (*h*) is used to compare the kinetics of GFP::NLS accumulation in the nucleus, which is indicative of the formation of a functional nuclear envelope. Time 0:00 corresponds to anaphase onset (AO). Time = min:s. *Scale bars* = 5 μm. *n* = number of cells. The mean ± 95% CI is presented for all scatter dot plots and graphs. A Mann-Whitney test was used to calculate *P* values (**** corresponds to $P < 0.0001$)

myosin enrichment at the cortex is detected upon partial depletion of Pbl activity. Second, myosin efflux is restored upon expression of a functional isoform of Pbl in *pbl*-depleted cells. While the mechanisms that target Ect2/Pbl to the midzone and subsequently to the equatorial membrane are well described[25, 38, 39], less is known about the control of Ect2/Pbl localization and function at the cortex of nascent cells. Similarly to human Ect2[25, 38], a pool of Pbl was detected briefly at the lateral and polar cortex during Nb cytokinesis (unpublished observation). Future experiments involving careful analysis of Pbl

dynamics after genetic and pharmacological perturbations will provide insight into the mechanism targeting Pbl to the cortex of the nascent daughter cells.

Some studies posit that the disassembly of components from the contractile ring contributes to its constriction[40–43]. Similarly, myosin efflux may be a mechanism for disassembling filaments from the ring during closure. Alternatively, mathematical models supported by mechanical studies have demonstrated that cleavage furrow ingression is unstable and that polar tension stabilizes the ring during closure[36, 44, 45]. Thus, myosin efflux may contribute

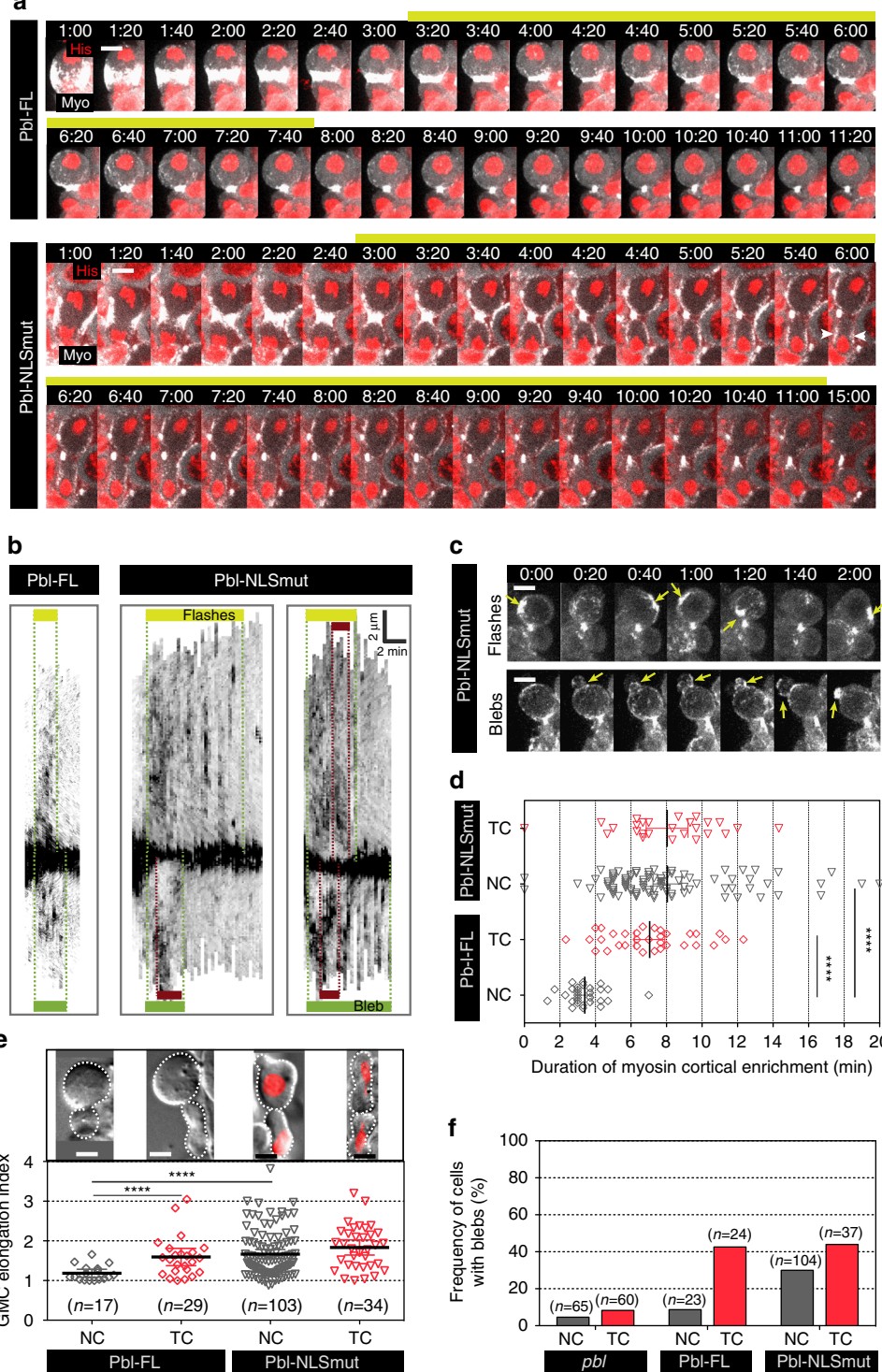

to increased cortical tension and cell shape stability during ring constriction. It is equally possible that myosin efflux serves as a ruler of daughter cell size, mediating subtle adjustment of ring position to ensure proper conservation of daughter cell size during symmetric or asymmetric division.

The presence of chromatids at the midzone causes the assembly of a large contractile ring enriched in myosin[15] (this study). We further showed that the wider the ring the more elongated the cell at the onset of furrowing. We reason that the contraction of a large ring exerts pressure on a wider area of the plasma membrane, thus inducing the first adaptive cell elongation. Studies have shown that during anaphase, the aurora B kinase relocates from the centromere to the midzone where it generates a gradient of phosphorylated species important for efficient cytokinesis and for the coordination of chromosome separation with NER[22, 46]. Concomitantly, chromatin-associated PP1 phosphatase activity counteracts aurora B kinase activity on chromatin, promoting NER and chromatin decondensation[47, 48]. We speculate that the presence of chromatin at the midzone perturbs the aurora B phosphorylation gradient due to the preservation of chromatin-associated PP1 phosphatase activity. Consequently, changes in the ratio of substrate phosphorylation:dephosphorylation subtly affect the assembly of the contractile ring. An alternative possibility is the persistence of an active chromatin-associated RCC1/Ran gradient at the midzone, which affects microtubule, actin, and myosin dynamics[49–55]. We can not rule out the possibility that the first adaptive elongation relies on the slight elongation of the spindle observed during the segregation of TCs. This spindle elongation may then affect the dynamics of myosin at the cortex during ring assembly.

Cell shape changes rely on the establishment of anisotropic cortical forces[56]. A well-studied example is the transition from a spherical to an elongated cell shape during anaphase, which precedes cytokinesis. This cell shape switch requires the depletion of actin and myosin filaments from the poles in parallel with their accumulation at the equator of the cell. The ensuing polar relaxation and equatorial tension triggers cell elongation[57–61]. Here we show that, in cells with TCs, the second wave of cell elongation relies on similar changes in cortical myosin distribution during myosin cortical enrichment. Upon myosin invasion of the cortex, it reorganizes into broad lateral rings. The pressure exerted on the cortex by the partial contraction of these lateral rings results in cell elongation. Interestingly, blebs frequently form in areas depleted of myosin (polar and adjacent to the ring) during this second phase of elongation. Blebs, which commonly protrude at the polar cortex during cytokinesis in human cell culture, are thought to act as something akin to cytoplasmic pressure valves that accommodate the increase in polar tension

during ring constriction[36, 61, 62]. Similarly, blebs that form during the second phase of elongation may release the increase in cytoplasmic pressure due to ectopic ring constriction.

We found that the segregation of TCs is not associated with extended spindle elongation. Therefore, it is unlikely that the enrichment of myosin on the lateral cortex results from changes in spindle morphology. We show, however, that the closer the TC is to the midzone, the more deformed the daughter cells. Conversely, the longer it takes for the TC arm to reintegrate into the main chromosome mass, the longer the cell remains elongated[15] (this study). This suggests that the presence and position of the TC relative to the midzone influences cortical myosin reorganization. We, and others, have shown that NER is specifically delayed in the presence of TCs[22, 24]. Importantly, previous studies attributed this delay to the midzone-associated aurora B gradient[22]. We speculate that the presence of TCs at the midzone induces an aurora B-dependent delay in NER, which favors the persistence of chromatin-associated cues that directly affect cortical protein dynamics. Importantly, studies have shown that chromatin-associated Sds22/PP1 modulates cortical protein dynamics by dephosphorylating the actin-binding protein moesin[58, 61].

However, our observation that the retention of Pbl in the cytoplasm in cells with normal chromatids is sufficient to maintain cortical myosin activity and to induce cell shape elongation similar to that observed in cells with TCs, suggests that chromatid-associated cues do not modulate myosin dynamics at the cortex directly. The fact that TCs trigger a delay in the reformation of a functional nuclear envelope raises the possibility that Pbl nuclear sequestration is also delayed, which, consequently, prolongs myosin activity at the cortex. The retention of Pbl in the cytoplasm may concomitantly promote the reorganization of cortical myosin as well as the assembly of new myosin filaments specifically at the lateral cortex.

It is important to note that a substantial amount of cells expressing Pbl-NLSmut with normal chromatids did not exhibit broad lateral myosin rings, but rather exhibited extensive blebbing and/or myosin flashes that we attribute to prolonged cortical myosin activity. This suggests that other cortical components that are normally sequestered in the nucleus at telophase facilitate cortical myosin reorganization into rings and subsequent cell elongation. A plausible candidate is anillin, since anillin interacts with myosin and is imported into the nucleus at the end of mitosis[63, 64]. It is equally possible that both chromatid-associated cues and prolonged cytoplasmic Pbl activity act in concert to promote myosin lateral ring formation required for the second elongation.

Several studies have identified mechanisms that facilitate the proper segregation of trailing or acentric

**Figure 6** The retention of Pbl in the cytoplasm during mitotic exit prolongs myosin activity at the polar cortex. **a** Time-lapse images of *pbl* mutant cells expressing Pbl full length (Pbl-FL) or a Pbl NLS mutant (Pbl-NLSmut) expressed from the endogenous promoter. The cells carry normal chromosomes and express Sqh::GFP (Myo, *gray*) and H2Az::mRFP (His, *red*). The *yellow bars* correspond to the time of myosin efflux in the Nb daughter cell. *White arrowheads* point to the position of ectopic myosin rings at the lateral regions of daughter cells. Time 0:00 corresponds to anaphase onset. See Supplementary Movie 5. **b** Kymographs of cortical myosin labeled with Sqh::GFP from one pole to the other over time for *pbl* mutant cells expressing Pbl-FL (one example) or Pbl-NLSmut (two examples) with NC. The first example illustrates a cell undergoing myosin flashes at the polar cortex (as shown in **c**). The second example shows a cell blebbing. The *yellow* and *green bars* show the duration of myosin cortical enrichment in Nb and GMC, respectively. The *x* and *y* axes of *scale bars* correspond to 2 min and 2 μm, respectively. **c** Time-lapse images of Pbl-NLSmut cells with NC expressing Sqh::GFP well after anaphase onset. The *top row* represents a cell where myosin flashes at the polar cortex and the *bottom row* shows a cell blebbing. *Yellow arrows* indicate myosin flashes (*top panels*) and blebs (*bottom panels*). **d** Scatter dot plot showing the duration of myosin cortical enrichment in *pbl* mutant cells expressing Pbl-FL (NC, *n* = 29; TC, *n* = 33) or Pbl-NLSmut (NC, *n* = 87; TC, *n* = 26). **e** Scatter dot plot showing the elongation index in *pbl* mutant cells expressing Pbl-FL or Pbl-NLSmut with NC or TC. The DIC images of cells expressing or not H2Az::mRFP (*red*) above the graph provides a representative example of each genotype at the time of elongation index measurement. A significant proportion of cells expressing Pbl-NLSmut elongates in the absence of trailing chromatids. **f** Frequency of blebs in *pbl* mutant cells expressing Pbl-FL or Pbl-NLSmut with NC or TC. Time = min:s. *n* = number of cells. *Scale bars* = 5 μm. The mean ± 95% CI is presented for all scatter dot plots and graphs. A Mann-Whitney test was used to calculate *P* values (**** corresponds to *P* < 0.0001)

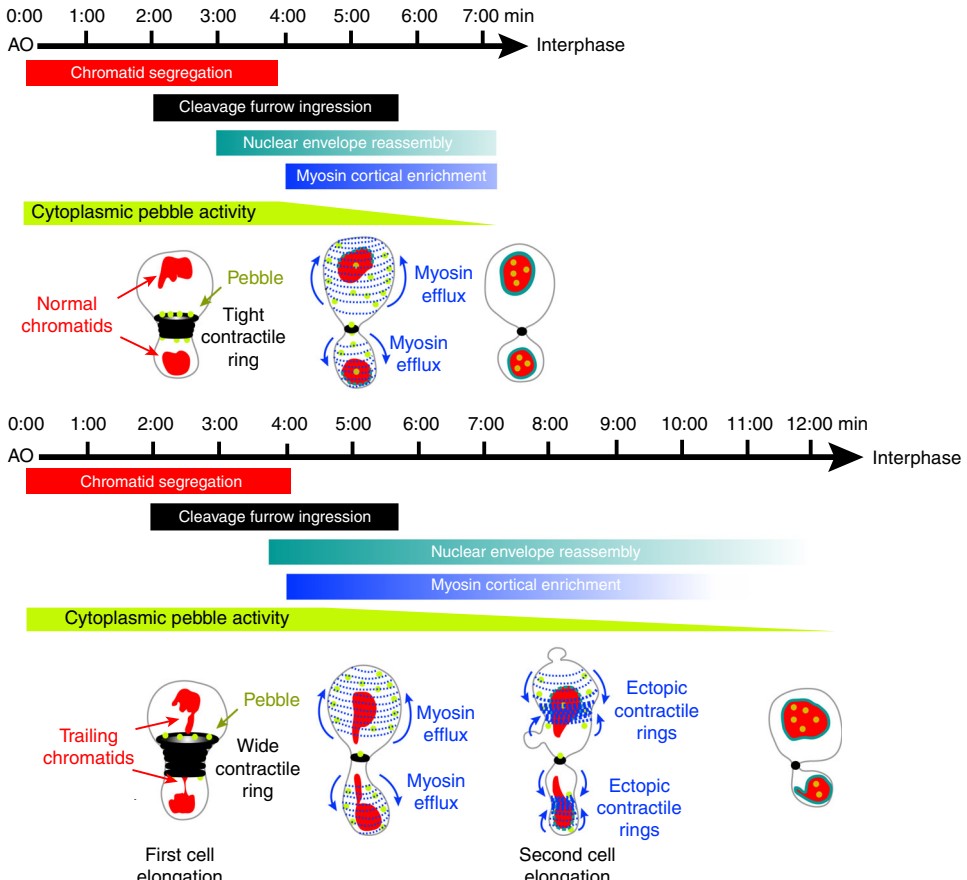

**Figure 7** Model for cell elongation in the presence of trailing chromatids. The *top and bottom panels* illustrate key mitotic exit events in cells with normal (*NC*) and trailing chromatids (*TC*), respectively, from anaphase onset (*AO*). The *red*, *black*, *cyan*, and *blue rectangles* represent the average duration of chromatids poleward movement, cleavage furrow ingression, nuclear envelope reassembly (*NER*), and myosin cortical enrichment, respectively. The *rectangles* are timely positioned with respect to AO. The variability in the duration of NER and myosin efflux is symbolized by *color fading* at the extremities. The *green shape* symbolizes putative pebble cytoplasmic activity throughout mitotic exit. The schemes illustrate the state of the four events listed above at key moments during mitotic exit. Three minutes after AO, Pbl (*green dots*) concentrates at the midzone where a tight contractile ring (*black circles*) assembles when chromatids (*red*) have segregated to the poles. The presence of trailing chromatids at the midzone favors the assembly of a wide ring, whose contraction generates the first cell elongation. Next, myosin initiates Pbl-mediated efflux and invades the polar cortex (*blue dashed thin curves*) while nuclear envelope starts reassembling on the chromosome mass at the poles (*cyan dashed lines*). In control cells, myosin disassembles from the polar cortex upon completion of NER (*cyan lines*), 7 min, on average, after AO. The presence of TC near the midzone induces a severe delay in NER completion. Consequently, the delay in Pbl nuclear import prolongs active cortical myosin, which reorganizes into ectopic lateral rings (*blue dashed thick curves*). The partial contraction of the rings promotes the second phase of elongation, allowing the clearance of the trailing chromatids from the cleavage plane. Myosin dissociates from the cortex upon completion of NER 12 min, on average, after AO

chromatids[6, 7, 11, 14, 15, 24, 65–68]. In this study, we provide novel insight into the mechanisms that promote cell elongation to coordinate chromatid segregation and cytokinesis. We discovered that TCs modulate myosin dynamics, first, during ring assembly and, then, during efflux. While the role of myosin efflux during cytokinesis remains to be defined, we found that it produces a cortical pool of active myosin at a critical time during chromosome segregation, where it can be rapidly employed to generate forces that facilitate the clearance of TCs from the midzone. We propose that the modulation of cortical myosin dynamics is part of the cellular response triggered by a "chromatid separation checkpoint" that delays NER when TCs are present at the midzone.

## Methods

**Drosophila stocks**. Flies were raised on maize meal media at 25 °C. The following lines: H2Az::mRFP ($P_{H2Az} >$ H2Az::mRFP1) and GFP::NLS ($P_{ubi-p63E} >$ GFP (S65T)::NLS) were provided by Bloomington (Betesda, USA). The transgenic stock carrying I-CreI was described previously[69]. sqh::GFP[70] and sqh::RFP stocks were provided by Roger Karess (Institut Jacques Monod, Paris, France). The stock carrying sqh::dendra2 was provided by Clemens Cabernard (University of Basel, Switzerland). The $pbl^5$ strong hypomorphic allele and the $pbl^3$ loss of function alleles were described previously[20]. $pbl^{MS}$, also called $pbl^{Z4836}$, was provided by Maurizio Gatti (University of Sapienza, Roma, Italy)[71]. The stock pbl > pbl-FL (86F8b) used to express Pbl-FL was provided by Michael Murray (University of Melbourne, Australia)[26]. Transgenic flies carrying pbl > pbl-NLSmut(86F8b) for Pbl-NLSmut expression were generated by BestGene, Inc. nup107 > GFP::Nup107 was described by ref. [72] and was provided by Valérie Doye (Jacques Monod Institute, Paris, France). $sqh^{AX3}$ was described by ref. [73]. The Gal4 line P{Gal4} neur[P72][74] was used to drive the expression of $P_{UAS} >$ H2A::RFP[75] in the sensory organ precursor cells. The stock C(2)EN; $P_{sqh} >$ Sqh::GFP40 was generated in the lab from the C(2)EN strain[76]. See Supplementary Table 1 for the complete genotype of the larvae, pupae, or embryo used in our experiments.

**Molecular biology**. The pebble genomic sequence described in ref. [26] was used to create Pbl-NLSmut. The Pbl NLS motif NKRKRKRFSQ was mutated to NAAAAAAFSQ (Genebridges). The resulting plasmid p[acman-pblNLSmut] was used to produce transgenic flies with the pbl-NLSmut sequence inserted at the landing site 86F8b on the third chromosome.

**Method for producing TCs**. Two methods for creating TCs have been described by ref. [15]. One method uses a fly strain carrying the chromosome C(2)EN[76], the other method uses the expression of the endonuclease I-CreI[15, 66, 67]. Third instar

larvae were heat shocked for 1 h at 37 °C in a water bath to induce the expression of the endonuclease I-CreI, and left to recover at room temperature for at least 1.5 h. The dividing Nbs were observed within 1-2 h after recovery.

**Live imaging of larval Nbs**. Third instar larval brains were dissected in phosphate-buffered saline (PBS) and slightly squashed between a slide and a coverslip by capillary forces[77]. The coverslip was sealed with halocarbon oil and the preparation was visualized immediately for a maximum period of 30 min.

**Live imaging of epithelial cells from pupae**. Pupal nota were prepared for live imaging 17 h post-puparium formation according to ref. [78]. The pupae were placed on double-stick tape with notum facing up. The cuticle was removed from head to notum and a coverslip coated with Voltalef 10S oil carefully placed onto pupae. Presumptive sensory organ precursor (SOP) pI cells were distinct from epidermal cells due to their selective expression of H2A::RFP.

**Live imaging of embryos**. Embryos were collected 3-4 h after deposition on grape-juice agar plates coated with yeast paste. They were dechorionated by hand, aligned on a coverslip, and coated with halocarbon oil[79].

**Immunostaining of Drosophila larval tissues**. Third instar larvae were dissected in PBS and digestive tracks, salivary glands, and central nervous system were fixed in PBS with 4% formaldehyde. Tissues were then rinsed in PBS with 0.1% triton and incubated with polyclonal anti-pebble guinea pig antibodies (GP14)[19] diluted at 1/1000 over night at 4 °C. After four washes in PBS with 0.1% triton, the preparations were incubated with Cy2-coupled anti-guinea antibodies (P0141, DAKO) diluted at 1/500. After four washes in PBS with 0.1% triton and a rinse in PBS, the tissues were mounted in Slowfade medium containing DAPI (Invitrogen) to visualize the DNA.

**Microscopy**. Live imaging of Nbs and pupae was performed at room temperature with either an Axio-observer inverted microscope (Carl Zeiss) equipped with a ×100 oil Plan-Apochromat objective lens (N/A 1.4), a spinning disk (CSUX-A1, Yokogawa), and an EMCCD Evolve camera (Princeton Instrument, Roper Scientific) or a Ti-DH inverted microscope (Nikon) equipped with a ×100 oil Plan-Apochromat objective lens (N/A 1.45), a spinning disk (CSU-W1-T1, Yokogawa), and a sCMOS camera (Roper Scientific). Images were acquired with Metamorph software (Molecular Device). Twelve to 20 Z of 0.5 µm steps were acquired every 20 s except in Fig. 3a–f, where one z plane was acquired every second (a-e) or every 5 sec (f). Fixed preparations and live embryos were observed using a DMI6000 inverted microscope equipped with a Plan-Apochromat ×63 objective (N/A 1.4) and a Leica TCS SP8 laser confocal imaging system. To induce the photoconversion of Sqh::Dendra2, cells were irradiated with a 405 nm laser (2.5% power, 1000 repetition) on a user-defined region for about 6–8 s using an Ilas2 system (Roper Scientific). Images in all figures are maximum projections except in Fig. 3a, b, d, e, f and DIC images.

**Image analysis and statistics**. Measurements were performed with ImageJ. The length of the contractile ring and total cell length presented in Fig. 1d, e, Fig. 4c and Supplementary Fig. 2b were measured at the onset of furrowing as illustrated in Fig. 1c. The contractile ring diameter measured over time and presented in Fig. 1h was normalized with the diameter of the cell at the onset of anaphase. Quantification of the myosin signal presented in Fig. 1f, g was performed by measuring the integrated density and the mean of the sqh::GFP signal at the contractile ring on one sagittal plane at the onset of furrowing. The quantification of the myosin signal presented in Figs. 2d and 3c was done by measuring the mean intensity of the sqh::GFP signal in three regions of the nascent GMC (Fig. 2d) or Nb (Fig. 3c) cortex over time from a max projection (Fig. 2d) and one sagittal plane (Fig. 3c). Nb and GMC elongation indices presented on Figs. 2h, j, 4g, 6e and Supplementary Fig. 2d were calculated as the length-to-width ratio as illustrated in Fig. 2g, at the maximal cell elongation, which is on average 6 min after anaphase onset. For Figs. 2d, k and 4i, the GMC elongation indices were calculated over time. For Supplementary Figs. 4c–f, spindle length and total cell length were normalized with the diameter of the cell at metaphase (c-d), and measured over time (c-d) or at maximal cell elongation (e-f) as shown in Supplementary Figs. 4b. To generate the kymographs of myosin dynamics at the cortex in Figs. 2c, 3b, 4e, and 6b and Supplementary Figs. 4a and 6c, we wrote a custom script in ImageJ. First, we drew a freehand line of 2 pixels thickness along the cortex from the apical Nb pole to the basal GMC pole on a single z plane at each time point starting 2 min after AO. The script converted the freehand lines onto straight lines centered on the contractile ring position. To calculate the rate of GFP::NLS nuclear accumulation presented in Fig. 5f, the GFP::NLS average intensity signal was quantified on a region of interest drawn at the presumptive nuclear position. We subtracted the average intensity of cytoplasmic GFP::NLS from this value. The data were normalized with the average intensity of GFP::NLS signal in the cytoplasm. For Figs. 1h and 5f, the data points were fit to a sigmoid curve ($Y = 1/(1 + 10^{((LogEC-X)*HillSlope)})$) using Prism (GraphPad) to determine the hillslope (h). All statistical analyses were performed with Prism software. A Mann-Whitney non-parametric test was used to calculate P values.

**Data availability**. All data generated or analyzed during this study are included in this published article (and its Supplementary Information Files) or available on request to authors.

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

## Acknowledgements

We thank Michael Murray (University of Melbourne, Australia), Michel Gho (Laboratoire de Biologie du developpement, Paris, France), alérie Doye (Institut Jacques Monod, Paris, France), Valérie Doye (Institut Jacques Monod, Paris, France), Roger Karess (Institute Jacques Monod, Paris, France) and Hugo Bellen (Duncan Neurological Research Institute, Houston, USA) for sharing their reagents and technical expertise. We also thank Derek McCusker (European Institute of Chemistry and Biology, Pessac, France) and all the team members for critical reading of the manuscript. E.M., L.B., and C.L. were supported by ANR-12-PDOC-0020-01 (ARC2-ChromSCeD). E.M. and C.L. were also supported by Centre National de la Recherche Scientifique (CNRS). M.-C.C. was supported by the University of Bordeaux. J.J. was supported by ERC-STG-2012 (GA311358-NoAneuploidy). A.T. and C.C. were supported by Swiss National Science Foundation. A.R. was supported by ERC-STG-2012 (GA311358-NoAneuploidy), Conseil Régional d'Aquitaine (20111301010), and Centre National de la Recherche Scientifique (CNRS).

## Author contributions

E.M., M.-C.C., .L.B., C.L., J.J., and A.R. performed the experiments. E.M. and A.R. designed the experiments and wrote the manuscript. A.R. supervised the work. A.T. and C.C. provided unpublished reagents.

## Additional information

**Competing interests:** The authors declare no competing financial interests.

