## [Peer Review File · Nature Communications]

Reviewers' Comments:

Reviewer #1 (Remarks to the Author)

In this paper Royou and colleagues provide insight on the mechanism of cell elongation in response to trailing/lagging chromatids. The authors report that this adaptive elongation involves the formation of a wider contractile ring during cytokinesis, followed by the flux of myosin from the ring to the polar cortex. This efflux is regulated by the Rho-GEF Pebble. The authors further report a delay in nuclear envelope reassembly in the presence of trailing chromatids. One key finding of this study is the observation that regulating the cytoplasmic/nuclear localization of Pebble is a critical step for this adaptive elongation response. All these findings are wrapped-up in a model in which spatial regulation of Pebble controls the extent of cellular elongation, particularly relevant in the presence of lagging chromatids. Overall, I find this work extremely elegant, with clever experiments and nicely executed. I am therefore in favor for publication and its wide dissemination. I do however have few minor comments/suggestions that I would like to draw the authors' attention to. A detailed critique follows below.

1- I find it unclear in which way the authors envision the modulation of polar myosin activity as part of an anaphase checkpoint. A checkpoint is a constitutive sensor that triggers a cell response. As so, I view the modulation of myosin activity as part of the cellular response triggered by the checkpoint that delays NER in response to lagging chromatids, not as part of the checkpoint itself. This should be clarified.

2- Page 4: the authors refer to the work of Afonso et al Science, 2015 in the context of a clearing mechanism before abscission, but this work did not claim anything regarding "prolonged condensation" or related this to cytokinesis (in fact, cytokinesis was experimentally inhibited due to the fact that S2 cells were grown in concanavalin A, which prevents furrow ingression). This should be corrected.

3- Page 8: the authors refer to the notion of an anaphase checkpoint that coordinates chromosome separation with NER and cite Afonso et al., Cell Cycle, 2014. I believe however that the appropriate references here should be Afonso et al., Science, 2015 and Maiato et al., Bioessays, 2015, in which the notion of a checkpoint was developed in full.

4- Page 9: the authors refer to the use of a GFP-NLS probe, but they do not provide any information about it in the methods.

5- Page 9: Etc2 should be Ect2.

6- Discussion: The authors refer to "accelerating the clearance of trailing chromatids". I find this confusing, do the trailing chromatids move more rapidly? Or have more time to move poleward?

7- The authors discuss the role of a putative Ran-GTP Gradient as a possible explanation for how trailing chromatids induce an adaptive elongation of the cell. This is in fact the only major weakness of this work, i.e. the authors fail to provide a link for how lagging chromosomes induce this adaptive elongation. Have the authors looked at the role of the Aurora B gradient? The fact that the cells with lagging chromatids assemble a wider contractile ring might change the organization of the Aurora B gradient at the spindle midzone and thus promote extended spindle elongation. In fact, one reason for the extended cell elongation might be the necessity to accommodate a bigger spindle, which might elongate further in response to changes in the Aurora B gradient. This might be worth investigating. While I recognize that this might be beyond the scope of this paper, it would be informative to provide at least the localization of Aurora B at the midzone in the presence of lagging chromatids (a simple IF in fixed material would suffice).

Related to this, it would be informative to provide data on the extent of spindle elongation in the presence of lagging chromosomes. Finally, given the previous implication of the Aurora B gradient in this potential anaphase checkpoint, I believe this would be worth discussing, at least in the same lines of the RanGTP gradient. The observation that cell elongation is controlled by the nuclear/cytoplasmic localization of Pebble, even in the absence of lagging chromosomes, strongly suggests that the signal for this adaptive cell elongation does not come from the chromosomes

themselves.

Helder Maiato

Reviewer #2 (Remarks to the Author)

For successful cell divisions that maintain the ploidy of cells, the timing of chromosome segregation and the physical cleavage of a single cell into two must be carefully coordinated. Lagging chromosomes and trailing chromatids present a challenge to this machinery as they position genetic material at the cleavage furrow, which if it closed upon them, could lead to the gain or loss of entire chromosomes and subsequent aneuploidy. There are chemical signals from the chromatids that can inhibit contractile ring closure, but the authors of this paper have previously described a mechanical solution to this problem called adaptive cell elongation that helps to clear trailing chromatids from the cleavage site by elongating daughter cells with a concomitant delay in ring closure. In this study, the authors demonstrate that this elongation is driven by the efflux of myosin from the contractile ring and that trailing chromatids stabilize the lateral bands of myosin. They observe the expected mechanical effects of this myosin redistribution including ectopic furrowing, increased blebbing and cell elongation. They show that trailing chromosomes cause a delay in nuclear envelope reassembly (NER) and that these activities are dependent upon the RhoGEF Ect2/Pbl. Finally, they show that blocking the import of Pbl into the nucleus drives cell elongation in the absence of trailing chromatids indicating that the sequestration of this Rho-activator is essential for shutting down this pathway.

Overall, I feel that the authors have described an interesting and important new cell behavior, which makes the publication of this report appropriate. The authors have been thorough in quantitating the behavior that they describe. The appropriate controls have been done, and the conclusions that they draw are all well-supported by the data that they have. Additionally, the Discussion anticipates the follow up questions that are raised in the minds of readers.

Minor Issues:

1. I would prefer that the authors not use the term polar to describe where the myosin accumulates. Polar has a very definite meaning to people in the yeast field and to most people who study cytokinesis. I would prefer the terms lateral or non-equatorial. It better conveys the localization and I have to confess to initially being somewhat confused when imagining how polar (again using the standard definition of a pole) myosin would promote rather than oppose cell elongation.
2. I found the hypothesis that MT nucleation from trailing chromatids is bolstering the thickness of the contractile ring to be far too speculative. One could also hypothesize that pools of chromatin bound RepoMan or Aurora B stimulate hyperassembly of myosin or centralspindlin. I would either include a broader discussion of possible hypotheses or the elimination of this point.
3. Although it is beyond the scope of this report, it would be very illuminating to see how Anillin and microtubules behave in this pathway. Given the transient nature of much of the lateral myosin, it would be interesting to know if there is coincident relocalization of Anillin away from the furrow and if MTs are interacting with these cortical sites. Additionally, it would be important to determine whether such elongation happens in cells in Anillin hypomorph mutants.
4. As the authors themselves point out, the evaluation of Pbl localization and dynamics at the furrow and on the lateral cortex will also be essential in the future.
5. Figure 1. The legend refers to blue arrowheads in 1a. However, there were no arrows present in the figure.
6. Page 4 "chromatids that remain at the midzone is thought" should be rewritten as "are thought".
7. Page 7 "was detected in two third of pbl" should be rewritten as "two thirds".
8. Page 9 Ect2 is written as Etc2.
9. Page 16 last sentence – chromatids should be plural.

Reviewer #3 (Remarks to the Author)

In their paper, Emilie and colleagues describe post-anaphase Myosin remodelling in neuroblasts that carry lagging damaged acentric chromosomes induced following the expression of the I-CreI endonuclease. This follows on from previous work of the lab, including an elegant recent study: "Cell elongation is an adaptive response for clearing long chromatid arms from the cleavage plane" in which they showed how aberrant chromosomal structures induced by the expression of I-CreI (which recruit AurB amongst other things) induce a change in the mode of cytokinesis. Cells are elongated, have a broad cytokinetic furrow, and exhibit ectopic furrowing. In this earlier paper they also argued that more Pbl is required for cell elongation than is required for ring formation.

In their new paper the team build on this by studying the later phases of cortical Myosin dynamics during neuroblast division in cells with lagging chromosomes (induced following I-CreI expression) in more detail. More specifically, they suggest that the period of high cortical Myosin activity post-ring closure is determined by the timing of Pbl sequestration in the nucleus, which is induced following nuclear envelope formation, a process that is delayed in cells with lagging chromosomes. Thus, aspects of the phenotype are recapitulated by a mutant variant of Pbl that lacks an NLS. While these observations might be considered subtleties of the system, the demonstration of the link between the timing of Pbl sequestration and nuclear envelope sealing is interesting and is likely to be a general feature of animal cell division.

General comments:

1. Myosin dynamics: The term efflux or outward flux are used to imply that Myosin flows along the cortex from the ring to the poles. The authors haven't shown that this is what happens. They have only shown that a proportion of the Myosin present in the ring makes it to the rest of the cortex later in the process. In fact, the Figures show that there is a depletion zone of Myosin close to the ring. This doesn't fit with a simple efflux model - and might suggest that cortical Myosin is incorporated into the ring at late stages (if this pool of Myosin turns over). Not enough has been done to determine the proportion of Myosin mini filaments in the ring that are disassembled before being re-accumulated at the rest of the cortex and to quantify Myosin flow along the cortex. Also the authors focus on some events while ignoring others: e.g. the timing of polar clearance of Myosin, the width of the central corset of Myosin prior to the onset of ring closure (which appears similar in wildtype cells and I-CreI cells), and the rate of ring closure (which appears slower in cells with lagging chromosomes). Finally, not enough has been done to determine the correlation between the timing of lagging chromosome movements and cortical activity.
2. The authors term the elongation an adaptation (even in the title). This is an overstatement. The authors haven't shown that the system is an evolutionary adaptation to deal with long chromosomes. It is not clear that the Pbl-induced changes in cortical mechanics helps.
3. Throughout the authors assume the defects they observe are due to trailing chromatids. In their original paper they looked at the C(2)EN chromosomes as a control for this. This led them to suggest that the changes in division induced by I-Cre1 expression are not the result of DNA damage/ DNA bridges, since cells that carry long C(2)EN chromosomes also appear elongated in anaphase. However, the work by the Maiato lab suggests a direct link between Aurora B, and envelope reformation. Thus, the team needs to revisit this by imaging Myosin and cell shape dynamics in this important control - especially as their previous study suggested that the extent of cell elongation was more pronounced in the I-Cre1 cells.
4. The authors separate their analysis of the cortex into a central region (where the Myosin ring forms) from the rest, which they deem the polar cortex. This is too crude a distinction. Again, the

authors fail to take into account the region with relatively low levels of Myosin at the region just outside of the ring when Myosin is accumulating more distally and moving towards the poles.

5. The really original parts of this paper relate to the timing of the nuclear re-accumulation of Pbl. However, the early Figures focus on much earlier events - that cannot be explained by this. In this the authors are repeating much of their previous analysis. Moreover, this is confusing given that there is no mechanistic link between the observation that Myosin re-accumulates at the cortex following ring closure and the failure to sequester Pbl in the nucleus much later. Note that if Pbl remains active later than usual it might, based on other literature, be expected to perturb abscission. Is this the case?

6. The conclusion section is much too speculative.

Bizarrely, it suggests that Myosin regulates NER not the other way around. The authors present no evidence for this. It is also suggested, without proper supporting evidence, that the extended period of Pbl activity in cells with lagging chromosomes accelerates their removal from the midzone and that myosin is part of an anaphase checkpoint.

It seems much more likely that:

- i) I-CreI defects lead to defects in anaphase segregation
- ii) Signals from the DNA delay closure and NER.
- iii) This prevents timely Pbl accumulation in the nucleus
- iv) This messes up cell shape.

To state otherwise requires evidence.

We would like to thank the reviewers for their positive and fair comments about our work and their suggestions that helped us improved our work. Described below is our detailed response to the reviewer comments.

Reviewer #1

Major comments:

1- I find it unclear in which way the authors envision the modulation of polar myosin activity as part of an anaphase checkpoint. A checkpoint is a constitutive sensor that triggers a cell response. As so, I view the modulation of myosin activity as part of the cellular response triggered by the checkpoint that delays NER in response to lagging chromatids, not as part of the checkpoint itself. This should be clarified.

In the abstract and last paragraph of the discussion we wrote: “the modulation of myosin efflux constitutes part of an anaphase checkpoint that delays NER to ensure proper transmission of the genetic material into daughter cells”. We agree that this sentence is poorly constructed and does not accurately convey what we meant to say. We did not intend to say that myosin is a component of the anaphase checkpoint but rather that myosin is the downstream effector of the anaphase checkpoint. We have changed the sentence in the abstract and last paragraph of the discussion by the one suggested by this reviewer: “the modulation of myosin activity is part of the cellular response triggered by “chromatid separation checkpoint” that delays NER in response to trailing chromatids”.

2- Page 4: the authors refer to the work of Afonso et al Science, 2015 in the context of a

clearing mechanism before abscission, but this work did not claim anything regarding “prolonged condensation” or related this to cytokinesis (in fact, cytokinesis was experimentally inhibited due to the fact that S2 cells were grown in concanavalin A, which prevents furrow ingression). This should be corrected.

In the revised manuscript, we have removed this reference from the statement and changed the sentence accordingly.

3- Page 8: the authors refer to the notion of an anaphase checkpoint that coordinates chromosome separation with NER and cite Afonso et al., *Cell Cycle*, 2014. I believe however that the appropriate references here should be Afonso et al., *Science*, 2015 and Maiato et al., *Bioessays*, 2015, in which the notion of a checkpoint was developed in full.

The citation Afonso et al., *Cell Cycle* 2014 has been changed for Afonso et al., *Science* 2014. We included Maiato et al., *Bioessays* 2015 as well in the new version.

4- Page 9: the authors refer to the use of a GFP-NLS probe, but they do not provide any information about it in the methods.

We did provide the source of the GFP-NLS stock in the material and methods; however, we called it p63E>GFP(S65T)::NLS instead of GFP::NLS. This has now been corrected.

7- The authors discuss the role of a putative Ran-GTP Gradient as a possible explanation for how trailing chromatids induce an adaptive elongation of the cell. This is in fact the only major weakness of this work, i.e. the authors fail to provide a link for how lagging chromosomes induce this adaptive elongation. Have the authors looked at the role of the Aurora B gradient? The fact that the cells with lagging chromatids assemble a wider contractile ring might change the organization of the Aurora B gradient at the spindle midzone and thus promote extended spindle elongation. In fact, one reason for the extended cell elongation might be the necessity to accommodate a bigger spindle, which might elongate further in response to changes in the Aurora B gradient. This might be worth investigating. While I recognize that this might be beyond the scope of this paper, it would be informative to provide at least the localization of Aurora B at the midzone in the presence of lagging chromatids (a simple IF in fixed material would suffice). Related to this, it would be informative to provide data on the extent of spindle elongation in the presence of lagging chromosomes. Finally, given the previous implication of the Aurora B gradient in this potential anaphase checkpoint, I believe this would be worth discussing, at least in the same lines of the RanGTP gradient. The observation that cell elongation is controlled by the nuclear/cytoplasmic localization of Pebble, even in the absence of lagging chromosomes, strongly suggests that the signal for this adaptive cell elongation does not come from the chromosomes themselves.

Dr Maiato is suggesting that cell elongation is occurring “to accommodate a bigger spindle”. In our previous study, we quantified the “extent of spindle elongation in the presence of lagging chromosomes”. We reported that the spindle slightly elongates in the presence of trailing chromatids but not to the extent of cell elongation (Kotadia et al., 2012, Fig. 2). We confirmed this result in this revised manuscript by (1) quantifying the spindle length over time for cells with NC and TC (shown in Fig. S3c), (2) quantifying the total cell

length over time in cells with NC and TC (shown in Fig. S3d), and (3) measuring the spindle length and total cell length when cells are at their maximal elongation (shown in Fig. S3e and f respectively). These results indicate that while the spindle elongates slightly but significantly in cells with TC, it never reaches the same length as the total cell length during elongation. The spindle elongates to a certain length and then stops, while the cell continues its elongation. This results in an increase in the distance between the spindle pole and the cortex (Kotadia et al. 2012, fig. 2). We further showed that *cnn* mutants containing reduced numbers of asters still undergo adaptive elongation (Kotadia et al., 2012, fig. S2). Collectively, these results suggest a minor role for microtubules in the mechanism of adaptive cell elongation. However, we cannot rule out the possibility that some spindle-associated forces are important for this process. We edited our discussion section to include the role of spindle elongation during the first adaptive cell elongation.

The reviewer suggested we look at Aurora B localization in the presence of lagging chromosomes. We feel that providing data on the localization of Aurora B in the presence of trailing chromatids without addressing the role of Aurora B in the mechanism of adaptive elongation would not be sufficiently informative to improve our current model. A thorough analysis of the role and dynamics of Aurora B during adaptive elongation is currently being investigated in the lab and requires a significant amount of work. We agree with the reviewer that this work is beyond the scope of this paper.

However, the discussion has been edited to include the potential role of the Aurora B gradient on myosin dynamics during adaptive elongation, as suggested by the reviewer.

Reviewer #2 (Remarks to the Author):

Minor Issues:

1. I would prefer that the authors not use the term polar to describe where the myosin accumulates. Polar has a very definite meaning to people in the yeast field and to most people who study cytokinesis. I would prefer the terms lateral or non-equatorial. It better conveys the localization and I have to confess to initially being somewhat confused when imagining how polar (again using the standard definition of a pole) myosin would promote rather than oppose cell elongation.

Based on this reviewer and reviewer #3 comments, we have changed the term polar cortex to either whole cortex, nascent daughter cell cortex, non-equatorial cortex or simply "cortical". We have provided additional data on myosin dynamics in three distinct regions of the daughter cell cortex, the region adjacent to the ring, the lateral region and the polar region. The quantification of myosin levels in these three cortical regions during myosin efflux and adaptive elongation is shown in fig. 3c and 2d respectively. See also our detailed response to point 1 of reviewer #3 below.

2. I found the hypothesis that MT nucleation from trailing chromatids is bolstering the thickness of the contractile ring to be far too speculative. One could also hypothesize that pools of chromatin bound RepoMan or Aurora B stimulate hyperassembly of myosin or centralspindlin. I would either include a broader discussion of possible hypotheses or the elimination of this point.

In our revised manuscript, we have edited our discussion to provide additional hypotheses

including the one suggested by this reviewer and reviewer #1 about the potential role of the Aurora B gradient as well as the chromatid-associated phosphatase activity on the modulation of myosin dynamics.

3. *Although it is beyond the scope of this report, it would be very illuminating to see how Anillin and microtubules behave in this pathway. Given the transient nature of much of the lateral myosin, it would be interesting to know if there is coincident relocalization of Anillin away from the furrow and if MTs are interacting with these cortical sites. Additionally, it would be important to determine whether such elongation happens in cells in Anillin hypomorph mutants.*

We agree with this reviewer that the analysis of Anillin dynamics and function during adaptive elongation is an interesting question to address. We have preliminary data supporting the idea that Anillin undergoes flux similar to myosin and is required for adaptive elongation. However, this work is an ongoing project in the lab and requires more thorough analyses.

4. *As the authors themselves point out, the evaluation of Pbl localization and dynamics at the furrow and on the lateral cortex will also be essential in the future.*

We agree with the reviewer and this is currently under investigation in the lab. We mentioned it in the text as unpublished data.

5. *Figure 1. The legend refers to blue arrowheads in 1a. However, there were no arrows present in the figure.*

This has been corrected.

6. *Page 4 “chromatids that remain at the midzone is thought” should be rewritten as “are thought”.*

This has been corrected.

7. *Page 7 “was detected in two third of pbl” should be rewritten as “two thirds”.*

This has been corrected.

8. *Page 9 Ect2 is written as Etc2.*

This has been corrected.

9. *Page 16 last sentence – chromatids should be plural.*

This has been corrected.

Reviewer #3 (Remarks to the Author):

General comments:

1. *Myosin dynamics: The term efflux or outward flux are used to imply that Myosin flows along the cortex from the ring to the poles. The authors haven't shown that this is what happens. They have only shown that a proportion of the Myosin present in the ring makes it to the rest of the cortex later in the process.*

We have provided additional evidence that a pool of myosin flows along the cortex from the contractile ring to the poles:

(1) An additional kymograph shows the dynamics of myosin from one pole to the other every second during efflux. This kymograph clearly shows the poleward flux of cortical myosin (Fig. 3b)

(2) We have quantified myosin levels over time at three cortical regions of the nascent daughter neuroblast, the region adjacent to the ring, the lateral region and the polar region. The graph shows high levels of myosin at the region adjacent to the ring at the onset of efflux then an increase in myosin levels at the lateral cortex and finally a prominent myosin signal at the polar cortex 2 minutes after efflux initiation (Fig. 3c).

In fact, the Figures show that there is a depletion zone of Myosin close to the ring.

We are not sure which figure the reviewer is referring to. The depletion zone close to the ring is evident in cells with trailing chromatids as shown in the kymograph of fig. 1g (labeled fig. 2c in our revised manuscript). In this revised manuscript we have documented more accurately the dynamics of myosin at the cortex during adaptive elongation by quantifying myosin levels at three distinct regions of the cortex (adjacent to the ring, lateral and polar). The graph shown in fig. 2d of the new version of the manuscript documents the depletion of myosin level from the region adjacent to the ring (red curve) and to some extent the poles (black curve) and the enrichment of myosin at the lateral cortex (brown curve) during adaptive cell elongation (blue curve).

This doesnt fit with a simple efflux model - and might suggest that cortical Myosin is incorporated into the ring at late stages (if this pool of Myosin turns over). Not enough has been done to determine the proportion of Myosin mini filaments in the ring that are disassembled before being re-accumulated at the rest of the cortex and to quantify Myosin flow along the cortex.

In this paper we reveal a novel feature of cytokinesis, myosin efflux that occurs in all cell types examined. In addition, we described how cortical myosin can be modulated to adapt the cell to the presence of trailing chromatids at the midzone. We acknowledge that myosin efflux and re-organization of cortical myosin during adaptive elongation are separate events. However, the latter depends on the former as our analysis of the *pbl* mutant cells show (*pbl* mutant cells do not exhibit efflux and do not undergo adaptive elongation in the presence of trailing chromatids).

The reviewer suggests that we, “*determine the proportion of Myosin mini filaments in the ring that are disassembled before being re-accumulated at the rest of the cortex*”. This is very challenging and we have not yet found a method that allows us to investigate this point with sufficient accuracy. However, we have added sentences in the revised version to include the possibility of a pool of myosin filaments assembling specifically on the lateral cortex during broad ring formation.

Also the authors focus on some events while ignoring others: e.g. the timing of polar clearance of Myosin, the width of the central corset of Myosin prior to the onset of ring closure (which appears similar in

wildtype cells and I-Cre1 cells), and the rate of ring closure (which appears slower in cells with lagging chromosomes). Finally, not enough has been done to determine the correlation between the timing of lagging chromosome movements and cortical activity.

We have provided additional data to address these requests.

1) Fig. S1 of this revised version provides the timing of polar myosin clearance, which is similar in cells with NC and TC. We did not measure the width of the corset as we are unsure how this information is relevant to this study.

2) We have included in fig. 1 a graph showing the rate of furrow invagination in cells that have assembled a wide ring in response to the presence of trailing chromatids. The rate of furrow invagination was slightly slower when cells started furrowing with a wide ring than normal cells (new fig. 1h). In addition, we provided the quantification of myosin levels in cells with NC with normal rings and TC with wide rings (fig. 1f and g). The total integrated density of myosin level was higher in wide rings than normal rings, which suggests an active enrichment of myosin at the contractile ring in the presence of lagging chromatids.

3) We have not determined the correlation between the timing of lagging chromosome movements and cortical activity. However, we have previously showed and confirmed in this revised manuscript that there is a correlation between the length of the trailing chromatid (which is inversely proportional to its rate of poleward movement) and the elongation index (new fig. 2i and j)(Kotadia et al., 2012, Fig. 1f). In addition, our previous study provides evidence that cells remain elongated until the trailing chromatid had reintegrated the chromosome mass (Kotadia et al., 2012, Fig. 1a and g). Finally, we show a correlation between the time it takes for the nuclear envelope to complete its reassembly (which depends on the time needed for the trailing chromatid to reintegrate the chromatin mass) and the duration of cortical activity (new fig. S5d).

2. The authors term the elongation an adaptation (even in the title). This is an overstatement. The authors haven't shown that the system is an evolutionary adaptation to deal with long chromosomes. It is not clear that the Pbl-induced changes in cortical mechanics helps.

In our previous report (Kotadia et al., 2012), we called this mechanism “cell elongation”. However, one reviewer of our previous study pointed out that the term “cell elongation” often refers to the switch in cell shape from spherical to oblong during anaphase B. To avoid confusion, the reviewer suggested that we rename the mechanism of cell elongation in response to trailing chromatids “adaptive elongation”. The term “adaptive elongation” refers to an adaptation of the cell to a stressful situation more than a mechanism that evolved amongst organisms.

3. Throughout the authors assume the the defects they observe are due to trailing chromatids. In their original paper they looked at the C(2)EN chromosomes as a control for this. This led them to suggest that the changes in division induced by I-Cre1 expression are not the result of DNA damage/ DNA bridges, since cells that carry long C(2)EN chromosomes also appear elongated in anaphase. However, the work by the Maiato lab suggests a direct link between Aurora B, and envelope reformation. Thus, the team needs to revisit this by imaging Myosin and cell shape dynamics in this important control - especially as their previous study suggested that the extent of cell elongation was more pronounced in the I-Cre1 cells.

If we understand correctly, the reviewer suggests that we study the dynamics of myosin in cells carrying a C(2)EN chromosome to control that adaptive elongation still occurs independently of DNA damage. Since, (1) we have previously provided evidence that cells carrying a C(2)EN, without DNA damage, still undergoes cell elongation (Kotadia et al., 2012, fig. S1), and (2) our new study demonstrates that cell elongation is driven by myosin cortical activity, we feel that we have provided compelling evidence that myosin-driven cell elongation is independent of the presence of DNA damage. Analyzing myosin dynamics in cells carrying a C(2)EN therefore seems beyond the scope of this study.

4. The authors separate their analysis of the cortex into a central region (where the Myosin ring forms) from the rest, which they deem the polar cortex. This is too crude a distinction. Again, the authors fail to take into account the region with relatively low levels of Myosin at the region just outside of the ring when Myosin is accumulating more distally and moving towards the poles.

See our detailed response above (comment 1. of reviewer #3).

5. The really original parts of this paper relate to the timing of the nuclear re-accumulation of Pbl. However, the early Figures focus on much earlier events - that cannot be explained by this. In this the authors are repeating much of their previous analysis. Moreover, this is confusing given that there is no mechanistic link between the observation that Myosin re-accumulates at the cortex following ring closure and the failure to sequester Pbl in the nucleus much later. Note that if Pbl remains active later than usual it might, based on other literature, be expected to perturb abscission. Is this the case?

In this study we have uncovered a novel feature of cytokinesis, myosin efflux, that occurs in various cell types. The signal that triggers myosin efflux is not known. However, the fact that a decrease in Pbl activity impairs myosin efflux suggests that Pbl activity is required for initiation of efflux. The signal that promotes myosin efflux as well as the biological function of myosin efflux is currently under investigation in the lab.

This reviewer is asking if abscission is perturbed during adaptive elongation. We did not study abscission in this report, as we feel that it is not within the scope of this paper. However, we do not observe an increase in multinucleated cells after I-Crel expression and in cells expression Pbl-NLSmut suggesting that abscission is not perturbed (data not shown).

6. The conclusion section is much too speculative.

Bizarrely, it suggests that Myosin regulates NER not the other way around. The authors present no evidence for this. It is also suggested, without proper supporting evidence, that the extended period of Pbl activity in cells with lagging chromosomes accelerates their removal from the midzone and that myosin is part of an anaphase checkpoint.

It seems much more likely that:

i) I-Crel defects lead to defects in anaphase segregation

ii) Signals from the DNA delay closure and NER.

iii) This prevents timely Pbl accumulation in the nucleus

iv) This messes up cell shape.

To state otherwise requires evidence.

We have addressed this point in the first comment to reviewer #1. Indeed, the last sentence of the abstract and the conclusion of the original submission was insufficiently clear. We did not intend to say that Myosin was the anaphase checkpoint that delays NER, but that the modulation of myosin cortical activity was the downstream effect of the anaphase checkpoint that delays NER in the presence of trailing chromatids.

Reviewers' Comments:

Reviewer #1:

Remarks to the Author:

The authors have addressed all my previous concerns and I can now recommend publication of this elegant work.

Reviewer #2:

Remarks to the Author:

I am satisfied with the changes that the authors made to the manuscript and figures and recommend it for publication.

Eric Griffis

Reviewer #3:

Remarks to the Author:

The revised paper and rebuttal address some of the comments raised in the reviews.

However, the authors have ignored many of the remarks made in the reviews.

This is a pity and it means that in my view the paper unnecessarily misrepresents interesting findings. If the authors addressed these points (as was suggested last time and as restated below) I would be happy to support its publication. This requires little in terms of experiments, just more care with language and interpretation of data.

While the review process can be annoying. It is good to treat the reviewers' comments in good faith. Reviewers are (I would hope) trying to help the authors publish a good paper that helps enlighten readers to a new piece of important science.

1. The authors have not changed anything in response to comments about their use of terms that don't present the data in the clearest light, i.e. they seem to intentionally or unintentionally misrepresent some of the findings.

a. EFFLUX

The analysis does not show Myosin efflux from the ring.

That data in 3b are very clear.

It shows that Myosin accumulates at the cortex, and that this accumulation proceeds from the cell centre to the cell pole. As pointed out previously, throughout this process, there is no accumulation of Myosin in the region next to the ring as would be expected in an efflux model. If there was efflux, material from the ring would spread out across the cortex (without a gap), visible as strong diagonal lines in 3b, and would be depleted from the ring as it moved. In addition, if efflux occurred the photo-conversion experiment (Figure 3d) wouldn't have led to the observed symmetric accumulation of Myosin along the cortex. Converted Myosin would only be present on one side. For these reasons, the term efflux therefore mis-represents the real picture. This doesn't mean the study isn't worth publishing in this journal, but the sloppy use of terms will only serve to mislead readers. The statement in the abstract "outward flux of myosin from the ring toward the polar cortex during ring constriction" also misrepresents the data.

b. The authors have not shown it this is an ADAPTIVE response. This term should therefore be removed. If a previous reviewer suggested this term be used, I think this previous reviewer was wrong. To show the response is adaptive rather than an epiphenomenon the authors need to show

that it aids cell division itself, i.e. that there are dire consequences for the partial pbl loss of function in TC cells. I suggest the authors remove the term OR better define the term to make it clear that they provide NO evidence for cortical flow of material from the ring out to cell poles.

c. The sentence in the abstract is still confusing. It suggests Myosin affects NER:

"We propose that the modulation of cortical myosin dynamics is part of the cellular response triggered by a "chromatid separation checkpoint" that delays NER when trailing chromatids are present at the mid zone."

better to state:

" We propose that the modulation of cortical myosin dynamics is part of the cellular response observed when lagging chromatids are present at the mid zone."

Why does this occur? Because the nucleus is slow to form!

2. The authors state in many places in their rebuttal that work suggested by reviewers is beyond the scope of this paper. This is a very weak argument in some cases.

a. It is important that they retest CEN(2)EN. This isnt beyond the scope. This is a control for this study. If they dont show this they need to explain very clearly in the text that DNA damage could explain the data.

b. The authors state that the "signal that promotes myosin efflux as well as the biological function of myosin efflux is currently under investigation in the lab."

This is the subject of the paper. If they have more compelling data to show its function, it should be shown if they want to claim its adaptive. Does it show that the process aids cell division?

c. The authors state that they didn't measure the width of the myosin cortex because they didn't think it relevant. This is important. It tests whether WHEN TC chromosomes induce a change in the width of the furrow. This is the subject of the paper and is relevant. In images it appears very similar in TC and normal cells. Is this the case? Its not difficult to measure.

3. Two reviewers asked about Aurora B.

I would suggest that the authors show a stain if they have the data as they suggest.

It will help to show readers what is happening.

Response to reviewer 3:

1. The authors have not changed anything in response to comments about their use of terms that don't present the data in the clearest light, i.e. they seem to intentionally or unintentionally misrepresent some of the findings.

a. EFFLUX

That data in 3b are very clear. The analysis does not show Myosin efflux from the ring. **there is no accumulation of Myosin in the region next to the ring as would be expected in an efflux model.**

We respectfully disagree with this reviewer's interpretation of our data in figure 3b. Below are panels extracted from figure 3b where the accumulation of myosin in the region next to the ring is observed 25 seconds after initiation of efflux. Subsequently, myosin accumulates at the lateral cortex and then reaches the polar cortex. Based on these observations, we use the terms outward flow and efflux in the text.

We now provide additional evidence that myosin undergoes efflux from the contractile ring. The new panels (d and e) in figure 3 show a surface view of a cell expressing Sqh::GFP at initiation of myosin efflux. Patches of myosin move out of the ring towards the pole, traversing the region near the ring. Reviewer 3 acknowledges that in an efflux model, myosin would enrich the region near the contractile ring. Since figures 3b, d and e show enrichment of myosin in the region near the contractile ring, we feel that the term efflux best illustrates these specific and conserved myosin dynamics at the cortex during cytokinesis.

However, we acknowledge that in some instances, our use of the term efflux may be more accurately referred to as myosin cortical enrichment. We have therefore thoroughly edited the text to replace “efflux” with “myosin cortical enrichment” when referring to myosin dynamics during cell elongation. For

instance, in figures 2f, 4f and 6d, we have now changed the axes labels from “duration of myosin efflux” to “duration of myosin cortical enrichment”. We have also stated in the text that we cannot exclude the possibility that a pool of cytoplasmic myosin contributes to the transient myosin cortical enrichment.

If there was efflux, material from the ring would spread out across the cortex (without a gap), visible as strong diagonal lines in 3b, and would be depleted from the ring as it moved.

We have now provided an additional movie of a surface view of a cell expressing Sqh::GFP during efflux initiation (figure 3d and e). This movie revealed that myosin forms patches that move from the ring towards the pole (see comment above). This explains the punctate appearance of myosin in figure 3b and the gap seen in the corresponding kymograph (3c). We are unsure why the reviewer feels that efflux would result in the uninterrupted outward flow of myosin; we see no reason why the pool of myosin undergoing efflux should not display heterogeneous diffusion on the plasma membrane that would generate the punctate pattern that we observe.

In addition, if efflux occurred the photo-conversion experiment (Figure 3d) wouldn't have led to the observed symmetric accumulation of Myosin along the cortex. Converted Myosin would only be present on one side. For these reasons, the term efflux therefore mis-represents the real picture. This doesn't mean the study isn't worth publishing in this journal, but the sloppy use of terms will only serve to mislead readers.

If we understand the reviewer's point correctly, they feel that the term efflux implies directed transport, rather than a diffusion-based process. We see no reason why the phenomenon that we are describing could not be referred to as efflux, since the definition of efflux is, "the flowing out of a substance". After photo-conversion, Sqh::dendra2 undergoes outward flow along the cortex (figure 3f). Moreover, myosin patches sometimes move diagonally towards the pole (figure 3d' and e).

The statement in the abstract "outward flux of myosin from the ring toward the polar cortex during ring constriction" also misrepresents the data.

We have edited the abstract to avoid misrepresentation of the data.

b. The authors have not shown it this is an ADAPTIVE response. This term should therefore be removed. If a previous reviewer suggested this term be used, I think this previous reviewer was wrong. To show the response is adaptive rather than an epiphenomenon the authors need to show that it aids cell

division itself, i.e. that there are dire consequences for the partial *pbl* loss of function in TC cells.

We have previously demonstrated the dire consequences to cellular viability resulting from the presence of trailing chromatids in a *pbl* mutant (Kotadia et al. 2012). We observed that the mutant displayed morphological defects such as rough eyes and a notched wing phenotype that is due to cell loss during development. We interpret the cell loss to be a consequence of aneuploidy due to the failure to trigger cell elongation. However, we have removed the term “adaptive cell elongation” and changed it to “cell elongation”.

c. The sentence in the abstract is still confusing. It suggests Myosin affects NER: "We propose that the modulation of cortical myosin dynamics is part of the cellular response triggered by a “chromatid separation checkpoint” that delays NER when trailing chromatids are present at the mid zone." better to state: " We propose that the modulation of cortical myosin dynamics is part of the cellular response observed when lagging chromatids are present at the mid zone." Why does this occur? Because the nucleus is slow to form!

Studies from reviewer 1’s lab have revealed the existence of a “chromosome separation checkpoint” that delays NER when chromatids are present at the midzone (Afonso et al., 2014). We have previously edited the sentence in question according to reviewer 1's suggestion.

2. The authors state in many places in their rebuttal that work suggested by reviewers is beyond the scope of this paper. This is a very weak argument in some cases.

a. It is important that they retest CEN(2)EN.

If adaptive elongation was triggered by the mitotic response to DNA damage that we uncovered (Royou et al., 2010; Derive et al, 2015), all cells expressing I-Crel would undergo adaptive elongation regardless of the length of the chromosomes. However, we demonstrated that the extent of elongation was correlated with the length of the trailing chromatid (Kotadia et al, 2012, this study, fig 2). In addition, the control experiments with C(2)EN have previously been published (Kotadia et al., 2012).

Nevertheless, as requested by this reviewer, we monitored myosin in cells carrying a C(2)EN (new figure S2). We found that the length of the ring at onset of furrowing is greater in cells carrying C(2)EN than NC cells. In addition, the duration of myosin cortical enrichment is prolonged in cells with the compound chromosome. Finally, the GMC elongation index is significantly higher in cells carrying the C(2)EN than cells with NC. These additional data,

in addition to the previously published C(2)EN experiment (Kotadia et al., 2012), demonstrate that cell elongation is not due to the DNA damage response.

Moreover, to further convince this reviewer that the cell elongation in the presence of I-Crel-induced trailing chromatids is not due to the BubR1 and Bub3-mediated DNA damage response, we have provided an additional figure to this rebuttal letter showing *rad50* mutant cells exhibiting severe chromosome segregation defects without BubR1 and Bub3 ectopic localization on chromosome arms (see annexe 1). These *rad50* cells undergo dramatic cell elongation in the presence of lagging chromosomes.

b. The authors state that the "signal that promotes myosin efflux as well as the biological function of myosin efflux is currently under investigation in the lab." This is the subject of the paper. If they have more compelling data to show its function, it should be shown if they want to claim its adaptive. Does it show that the process aids cell division?

There appears to be a mis-understanding with our use of the word "adaptive". We used the term "adaptive elongation" to illustrate the cellular response to the presence of trailing chromatids at the midzone. The focus of this work was to determine the molecular mechanism by which myosin mediates cell elongation to facilitate the clearance of trailing chromatids from the midzone. In doing so, we uncovered a novel feature of cytokinesis: myosin efflux. The term "efflux" refers specifically to the transient flow of myosin from the ring to the pole during cytokinesis in normal cells and cells with trailing chromatids, which we thoroughly describe in this work. These myosin cortical dynamics are a new feature of cytokinesis and raise exciting questions for the future: what is the biological function of myosin efflux and what controls its initiation. One possible function for myosin efflux that we stated in the discussion is that "it produces a cortical pool of active myosin at a critical time during chromosome segregation, where it can be rapidly employed to generate forces that facilitate the clearance of trailing chromatids from the midzone". We also showed that full Pbl activity is required for myosin efflux initiation. Finally, myosin dissociation from the cortex is dependent upon Pbl sequestration in the nucleus, which depends on the re-assembly of a functional nuclear envelope. We feel that these are important findings that will generate interest in the field and lead to additional studies on the underlying molecular mechanism. However, the detailed molecular pathway triggering myosin efflux is not fully elucidated here and additional investigation will be required.

c. The authors state that they didn't measure the width of the myosin cortex

because they didn't think it relevant. This is important. It tests whether WHEN TC chromosomes induce a change in the width of the furrow. This is the subject of the paper and is relevant. In images it appears very similar in TC and normal cells. Is this the case? Its not difficult to measure.

During the first review, we did not understand the reason for measuring “the width of the myosin corset” as originally requested by reviewer 3, since the rationale of the reviewer was not explained. Now that this has been clarified, we have provided a graph showing this quantification in figure S1b. This quantification shows that, in cells with trailing chromatids exhibiting wide rings at the onset of furrowing, the rate of ring compression (length of the ring over time) stalls at the onset of furrowing until the initiation of efflux. Once efflux started, the length of the ring decreased dramatically. We are grateful for this suggestion.

3. Two reviewers asked about Aurora B.

I would suggest that the authors show a stain if they have the data as they suggest. It will help to show readers what is happening.

With respect, we feel that showing Aurora B staining without addressing the role of Aurora B during the segregation of trailing chromatids will not help the readers to see what is happening. Reviewer 1, who originally asked for this experiment, agreed with our statement that "providing data on the localization of Aurora B in the presence of trailing chromatids without addressing the role of Aurora B in the mechanism of adaptive elongation would not be sufficiently informative to improve our current model".

Annexe 1

a

b

c

(a) Time-lapse of *rad50* mutant neuroblasts expressing Sqh::GFP (Myo, grey) and H2Az::mRFP (His, Red). Time= min:sec. (b) Time-lapse of *rad50* mutant neuroblast expressing GFP::BubR1 (BuBR1, grey) and H2Az::mRFP (His, red) . The cyan arrows point to the kinetochore localisation of BubR1. (c) Time-lapse of *rad50* mutant neuroblast expressing GFP::Bub3 (Bub3, grey) and H2Az::mRFP (His, red) . The cyan arrows point to the kinetochore localisation of Bub3. The dashed white lines delineate the cells. Time=min:sec

Reviewers' Comments:

Reviewer #3:

Remarks to the Author:

The authors have now addressed all the questions I raised in my reviews.

The revised paper explains the data in a clear and compelling manner.

In addition, the new data are great. I particularly, like 3d/e!!!

The paper will be an excellent addition to the literature.

I look forward to seeing it in print!

Minor errors:

"This was accompanied with a brief stall in the rate of ring compression (Supplementary Fig. 1b)."

This could be said more clearly, e.g., "rate at which the central band of Myosin collapses to a ring is delayed in TCs."

This is a striking observation as is clear from 1a/b.

Also, there is an error in the label.

S1b "Ring lcompression " error.

p6 "As observed previously, the extend of cell elongation was"
should be "extent"

Response to reviewer 3:

Minor errors:

"This was accompanied with a brief stall in the rate of ring compression (Supplementary Fig. 1b)."

This could be said more clearly, e.g., "rate at which the central band of Myosin collapses to a ring is delayed in TCs."

We have changed the sentence accordingly.

Also, there is an error in the label.

S1b "Ring lcompression " error.

This has been corrected.

p6 "As observed previously, the extend of cell elongation was" should be "extent"

This has been corrected.